# Adaptation of *Drosophila* larva foraging in response to changes in food resources

**Marina E Wosniack[1], Dylan Festa[2], Nan Hu[3], Julijana Gjorgjieva[1,2]\*, Jimena Berni[3,4]\***

[1]Computation in Neural Circuits Group, Max Planck Institute for Brain Research, Frankfurt, Germany; [2]School of Life Sciences, Technical University of Munich, Munich, Germany; [3]Department of Zoology, University of Cambridge, Cambridge, United Kingdom; [4]Brighton and Sussex Medical School,, University of Sussex, Brighton, United Kingdom

**Abstract** All animals face the challenge of finding nutritious resources in a changing environment. To maximize lifetime fitness, the exploratory behavior has to be flexible, but which behavioral elements adapt and what triggers those changes remain elusive. Using experiments and modeling, we characterized extensively how *Drosophila* larvae foraging adapts to different food quality and distribution and how the foraging genetic background influences this adaptation. Our work shows that different food properties modulated specific motor programs. Food quality controls the traveled distance by modulating crawling speed and frequency of pauses and turns. Food distribution, and in particular the food–no food interface, controls turning behavior, stimulating turns toward the food when reaching the patch border and increasing the proportion of time spent within patches of food. Finally, the polymorphism in the *foraging* gene (rover–sitter) of the larvae adjusts the magnitude of the behavioral response to different food conditions. This study defines several levels of control of foraging and provides the basis for the systematic identification of the neuronal circuits and mechanisms controlling each behavioral response.

**\*For correspondence:**
gjorgjieva@tum.de (JG);
j.berni@sussex.ac.uk (JB)

## Editor's evaluation

This paper contributes to the growing body of literature that investigates foraging in complex sensory landscapes. It is therefore of interest to both neuroscientists and ecologists. Using behavioral analysis and computational modeling, the authors characterize different behavioral components of the foraging strategy adopted by the *Drosophila* larva as a function of food quality and food distribution. Altogether, this works sets the stage for investigating the genetic and neural-circuit bases underlying the control of foraging behavior.

## Introduction

Most moving organisms need to explore their surroundings to increase their chances of finding nutritious resources. This is a challenging task in natural environments, where food quality varies both in time (e.g., seasonal effects) and space (e.g., patchy distribution). Therefore, the exploratory behavior of animals has to be flexible and adapt to environmental challenges. From the perspective of evolutionary ecology, foraging strategies have evolved to maximize lifetime fitness under distinct constraints (*Stephens and Krebs, 1987*) including the concentration of food inside patches (*Charnov, 1976*). Accordingly, several hypotheses and models have been developed to predict the optimal foraging strategy that an animal will adopt (*Stephens and Charnov, 1982*; *Viswanathan et al., 2011*). These models postulate that animals will use different strategies depending on the

distribution of the resources. In environments where resources are abundant, animals will search and exploit them performing short movements in random directions, in patterns well approximated by Brownian random walks. When resources are sparse, and foragers have incomplete knowledge about their location, a more diffusive strategy is needed, with an alternation between short- and long-range movements, which can be modeled as a Lévy random walk. Analysis of animal movements in the wild has demonstrated that environmental context can induce the switch between Lévy to Brownian movement patterns (*Humphries et al., 2010*), but the effective mechanisms behind the implementation of such behavior (e.g., cognitive capacity, memory) often remain elusive (*Budaev et al., 2019*). Understanding the motor mechanisms that regulate the execution of different movement strategies and the transitions between them could provide insight into how the nervous system can drive the search for resources in complex and ever-changing environments. *Drosophila* larva is an excellent model to study this question, because the movement of single animals can be tracked for long periods of time in a controlled environment.

Larvae of the fruit-fly are constantly foraging and feeding to fulfill their nutritional needs for the following non-feeding pupal stage. They explore the substrate by executing sequences of crawls, pauses, and turns (*Berni, 2015*; *Berni et al., 2012*) and can efficiently explore an environment even without brain input (*Sims et al., 2019*). Larvae approach (or avoid) sources of odor by triggering oriented turns during chemotaxis (*Gomez-Marin et al., 2011*) and can also navigate through gradients of light intensity (*Kane et al., 2013*; *Humberg and Sprecher, 2018*), temperature (*Luo et al., 2010*; *Lahiri et al., 2011*), and mechanosensory cues (*Jovanic et al., 2019*). Their natural habitat is decaying vegetable matter distributed in patches (*Ringo, 2018*), and due to food decay and intraspecific competition larvae are constantly deciding what patch to visit and how long to stay before exploring for new higher quality food patches. This constant exploration comes at a high energetic cost since crawling behavior is very demanding (*Berrigan and Lighton, 1993*; *Berrigan and Pepin, 1995*).

The foraging behavior of *Drosophila* both in the larval and adult stages is influenced by the *foraging* (*for*) gene (*Sokolowski, 2001*; *Sokolowski et al., 1997*). Larvae with the rover allele crawl significantly longer paths on a yeast paste than larvae with the sitter allele, and a proportion of 70% rovers and 30% sitters is observed in natural populations (*Sokolowski, 2001*). Due to the higher dispersal of rover larvae, their pupae are usually found in the ground while those from sitter are usually found on the fruit (*Sokolowski et al., 1986*). However, it is not known if the behavioral differences between rover and sitter larvae can be observed in food substrates of different compositions, nor how rovers and sitters behave in a patchy environment of regions with and without food (even though it has been hypothesized that rover larvae are more likely than sitter to leave a patch of food to search for a new one, *Sokolowski, 2001*).

Previous studies on larval foraging focused on the behavior in homogeneous substrates, where larvae engage in a highly exploratory movement pattern if no food is available (*Berni et al., 2012*; *Godoy-Herrera et al., 1984*; *Sims et al., 2019*). However, the natural habitat of larvae is very patchy and it is not clear how they select feeding vs. exploring when the environment has food patches separated by areas without food. Previous studies have shown that larvae are more willing to leave a patch if the protein concentration is low but tend to stay in the patch if its nutritional content is adequate (*Ringo, 2018*). Nevertheless, these studies lack an individualized tracking of the path executed by larvae during patchy exploration.

Here, we investigate the mechanisms of foraging that adapt to changes in food distribution. To address this challenge, we investigate how (1) the quality of the food and (2) its distribution, homogenous vs. constrained in small patches, influence larval foraging. We test the effect of the rover and sitter genetic dimorphism in the different food distributions and disentangle the role of olfaction in remaining in food patches using anosmic animals. By combining a detailed analysis of individual larval trajectories from behavioral experiments and computational modeling, we characterize the elements of the navigation routine and show how they adapt to a changing environment. Our results show a modular adaptation to different food characteristics. Food quality modulates crawling speed, turning frequency, and fraction of pauses controlling the distance traveled and therefore the area explored. The patchy distribution of food triggers oriented turns toward the food at the patch interface, increasing the time larvae exploit the food inside the patch. The foraging polymorphism of rovers and sitters adjusts the degree of the behavioral response to different food conditions. The

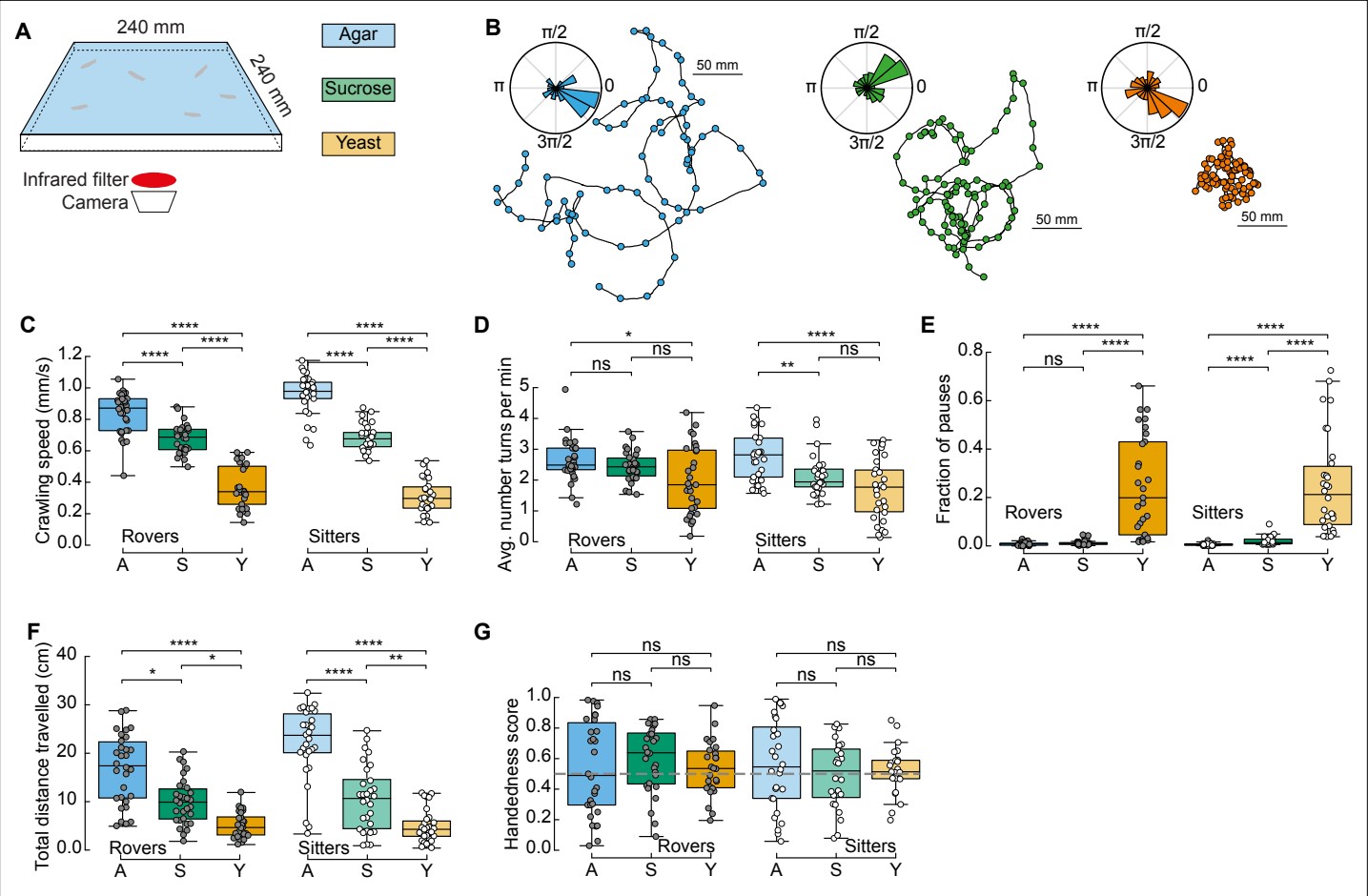

**Figure 1.** *Drosophila* larva exploratory behavior in homogeneous substrates. (**A**) Experimental setup: 10 larvae of the same phenotype (rover or sitter) were placed on the top of an agar-coated arena and recorded for 50 min, experiments were repeated three times with independent samples. Three types of substrates were used: agar-only (blue), sucrose (green), and yeast (orange). (**B**) Sample trajectories of rover larvae in the different substrates (top: agar, bottom left: sucrose, bottom right: yeast) with turning points identified by the RDP algorithm. Corresponding turning angle distributions are shown as an inset. (**C**) Average crawling speeds of rovers ($N = 30, 30, 29$) and sitters ($N = 29, 30, 30$) in the different substrates: agar (A, blue), sucrose (S, green), and yeast (Y, orange). The speed was calculated during bouts of crawls. Horizontal line indicates median, the box is drawn between the 25th and 75th percentiles, whiskers extend above and below the box to the most extreme data points within 1.5 times the interquartile range, points (gray for rovers, white for sitters) indicate all data points. (**D**) Average number of turns per minute registered in each trajectory. (**E**) Fraction of time in which larvae did not move (pauses). (**F**) Total distance traveled in 50 min. (**G**) Handedness score. The horizontal dashed line corresponds to a score of 0.5, that is, an equal number of counter-clockwise (CCW) and clockwise (CW) turns. Mann–Whitney–Wilcoxon test with Bonferroni correction was performed since the data were not normally distributed. ns: $0.05 < p < 1$, *$0.01 < p < 0.05$, **$0.001 < p < 0.01$, ****$p < 0.0001$. The number of larvae tested is detailed in *Table 1*. Statistical power and Cohen's size effect of non-significant comparisons are included in *Table 4*.

The online version of this article includes the following figure supplement(s) for figure 1:

**Figure supplement 1.** Comparison between rover and sitter behavior in different substrates.

detailed description of the larval behavior and the model presented here provide the basis for the systematic identification of the neuronal circuits and mechanisms controlling each behavioral response modulated by different food resources.

## Results

### Food quality controls the distance traveled modulating the speed and the frequency of pauses

To study the effect of different food substrates in foraging larvae, we devised a behavioral assay where larvae explore different substrates with minimal external stimuli (*Figure 1A*). The three different

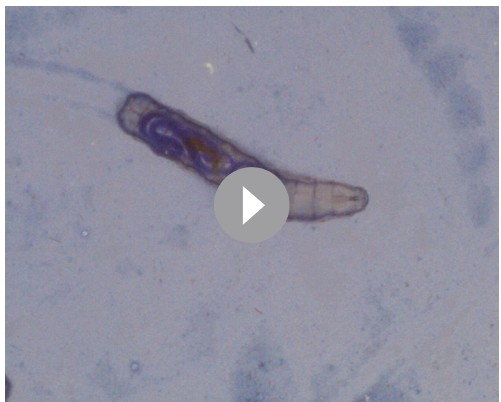

**Video 1.** Video of a larva pausing. A sitter larva was allowed to feed on a fine layer of yeast supplemented with 0.1% Bromophenol blue sodium salt (B5525, Sigma). During pauses the larvae are immobile and only the movement of their gut content can be seen. https://elifesciences.org/articles/75826/figures#video1

substrates (agar, sucrose, yeast) had the same agar density but distinct nutritional quality (with yeast being the richest due to its high content of protein) (Materials and methods). Wildtype larvae from different polymorphisms – rovers and sitters – were separately recorded because of previously reported differences in foraging behavior (*Sokolowski et al., 1997*). We recorded the free exploratory behavior of groups of 10 third-instar larvae in large arenas (240 × 240 mm²) for 50 min and then tracked each individual trajectory (*Risse et al., 2013*; *Sims et al., 2019*). Three independent replicates were analyzed. To identify salient turning points in the trajectory and to obtain the distribution of turning angles of each larva, we used the Ramer–Douglas–Peucker algorithm (Materials and methods). Larvae explored the three different substrates (*Figure 1B* and *Figure 1—figure supplement 1A*) by executing sequences of crawls, turns (marked as circles in the trajectories), and pauses. Interestingly, we observed that a preferential orientation – clockwise (CW) or counter-clockwise (CCW) – is present in many trajectories, and the paths described often have circular shapes (*Figure 1B*).

We found that the presence of food in the substrate had a strong effect on the larval crawling speed. Rover (sitter) larvae crawl on average at a speed of 0.84, 0.68, and 0.37 mm/s (0.96, 0.68, and 0.31 mm/s) in the agar, sucrose, and yeast, respectively (*Figure 1C*). In addition to changing speed, larvae suppressed turning in the food substrates, with rover (sitter) larvae executing an average of 2.65, 2.44, and 2.00 (2.79, 2.13, and 1.71) turns per minute in the agar, sucrose, and yeast, respectively (*Figure 1D*). Larvae also paused more often in the yeast substrate (*Figure 1E* and *Video 1*). Most pausing larvae were completely still, except for internal movements in their gut, suggesting they were digesting (*Video 1*). As a consequence, the total distance traveled showed a clear dependence with food quality, with yeast, the most nutritious food, generating the shorter path and consequently often a smaller explored area (*Figure 1B, F* and S1A).

We next quantified the individual orientation preference of each larva based on its turning angle distributions. We defined the handedness score *H* of a larva as the number of CCW turns divided by the total number of turns in the trajectory, that is, CCW and CW combined. Larvae with *H* > 0.5 (*H* < 0.5) have a bias to turn CCW (CW). Surprisingly, in both rover and sitter populations we found larvae with a very strong handedness, meaning that larvae have individual biases when turning in homogeneous environments that do not provide orientation cues in the form of sensory stimuli (*Figure 1G*).

Finally, we contrasted the differences in exploratory behavior of rovers and sitters in the different homogeneous substrates (*Figure 1—figure supplement 1B–G*). In particular, we were interested in evaluating if sitter larvae crawled significantly less than rovers in the first 5 min of the recording in the food substrates, as previously observed in experiments using yeast substrates (*Sokolowski, 1980*). We did not find significant differences between the crawled distances of rovers and sitters in the substrates that we tested. Thus, when the resources are distributed homogenously, the genetic foraging dimorphism could not be detected.

In summary, we have provided a detailed characterization of larval foraging behavior in homogenous substrates with different types of food. We found that larval crawling speed and probabilities to turn and to pause are behavioral elements that are adapted according to the quality of food.

## A phenomenological model of crawling describes larval exploratory behavior in patchy substrates

In ecological conditions, the fruit on which *Drosophila* eggs are laid and on which the larvae forage decays over time. To maximize their survival chances, and reduce competition, larvae therefore move

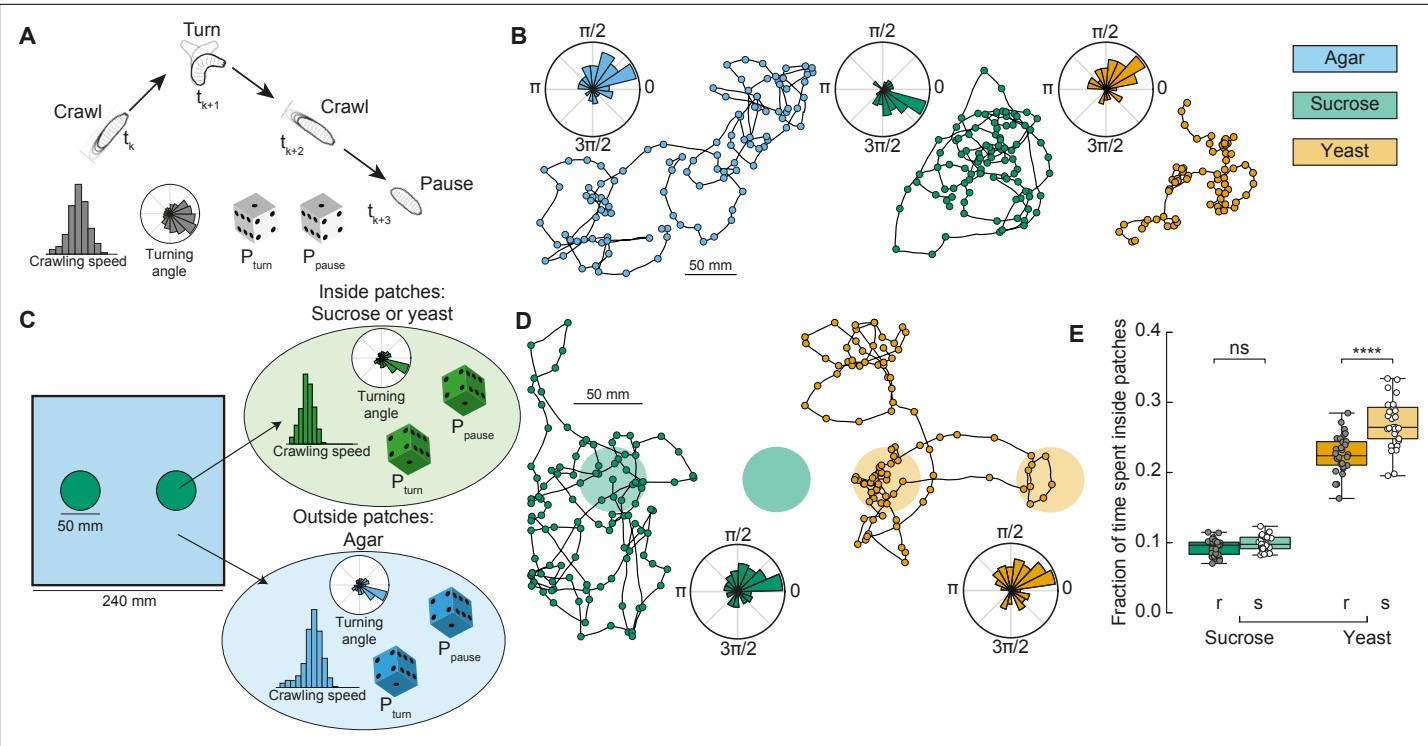

**Figure 2.** Model of larva crawling in different substrates. (**A**) Simulated larva crawls at time steps $t_k$ and $t_{k+2}$, turns at $t_{k+1}$, and makes a pause at $t_{k+3}$. Crawling speed and turning angle are sampled from normal and von Mises probability distributions, respectively. At each time step, there is a constant probability to turn $P_{turn}$ or to pause $P_{pause}$. (**B**) Sample model trajectories and turning angle distributions of sitter larvae simulated in different homogeneous substrates: agar (left), sucrose (middle), and yeast (right). (**C**) Simulations with patchy environments: food (sucrose or yeast) is distributed inside two circular regions, with agar in the remaining substrate. Crawling speeds, turning and pause probabilities are sampled from different distributions when the simulated larva is inside (green) or outside (blue) the patch. (**D**) Sample model trajectories and turning angle distributions of sitter larvae in simulated patchy substrates: sucrose (left) and yeast (right) patches. (**E**) Average fraction of time each simulated larva (rover (r), sitter (s), N = 30) spent inside patches (sucrose and yeast) in the simulations. Horizontal line indicates median, the box is drawn between the 25th and 75th percentiles, whiskers extend above and below the box to the most extreme data points that are within a distance to the box equal to 1.5 times the interquartile range and points indicate all data points. Mann–Whitney–Wilcoxon paired test two-sided. ns: 0.05 < p < 1, ****p < 0.0001.

The online version of this article includes the following figure supplement(s) for figure 2:

**Figure supplement 1.** Fraction of time spent inside patches.

toward food patches that are more nutritious and less crowded (*Del Pino et al., 2015*). Here, we designed a phenomenological model to simulate larval exploratory trajectories in different substrates based on our collected data (*Figure 1* and Methods). The model predicted the fraction of time larvae spent inside patches of food, as a measure of food exploitation, if larvae only used the information about the substrate while foraging. Each type of larva (rover, sitter) had a distribution of crawling speeds $v$ and probabilities to crawl $P_{crawl}$, to turn $P_{turn}$, and to pause $P_{pause}$ in a given time step for each type of homogeneous substrate: agar, sucrose, and yeast (*Figure 2A*). To capture the variability in the turning behavior, each simulated larva had its own set of parameters for the turning angle distribution based on a single recorded larva. The simulated trajectories preserved the CW or CCW orientation inherited from the turning angle distributions characterized in the experiments (*Figure 2B*).

Using our model based on crawling behavior in homogeneous substrates, we next tested how changes in the food distribution influence the exploratory trajectories of rovers and sitters. We modeled heterogeneous environments with two circular patches of food substrate with agar substrate in the rest of the arena (*Figure 2C*, see Materials and methods). The two patches had a fixed radius (25 mm) that corresponds to the surface area of a grape (*Xie et al., 2018*). We simulated larval exploration of rovers and sitters in patches of two food substrates – sucrose and yeast (*Figure 2D*). The initial position was picked at random in each simulation, but always inside one of the two food patches to match the experiments.

We next quantified the fraction of simulation time that rovers and sitters spent inside patches. For each larva, this was averaged over 30 simulation runs (*Figure 2E* and *Figure 2—figure supplement 1A, B*). Inside sucrose patches, the percentage of time spent inside patches was small for both rovers and sitters (9.2% and 9.8%, respectively). These values were only slightly larger than those in a simulated environment with patches made of agar (7.47% for rovers and 6.99% for sitters – *Figure 2—figure supplement 1A, C, D*) – that is, the same speed and probabilities to turn and pause inside and outside patches. This result is unsurprising since in our homogeneous substrate experiments with rover and sitter larvae both had similar behavior in the agar and sucrose arenas. In simulations with yeast patches, the percentage of time spent inside patches was higher for both rovers (22.6%) and sitters (26.9%). This increase can be linked to the slower speeds and more frequent pauses in the homogeneous yeast substrate executed by the larvae. In spite of non-significant differences in the crawling of rovers and sitters in the homogeneous yeast substrate (*Figure 1—figure supplement 1B–G*), in our model simulated sitter larvae remained longer inside the yeast patches due to their lower (though not significantly different average crawling speed in the homogeneous yeast substrate experiments).

Thus far, our model predicts that, in patchy environments, larvae spend a relatively small proportion of time inside patches (approximately 1% for sucrose and 3% for yeast) while exploring takes up most of their time with a significant energy cost. However, our model does not integrate other possible mechanisms that a larva might employ to remain inside a food patch besides decreasing its crawling speed and increasing the fraction of pause events. We therefore compared the model predictions on foraging efficiency in patchy environments with behavioral experiments.

## Increased proportion of time in patches relies on turns toward the patch center at the food–no food interface

We next recorded the larval behavior in arenas with patchy substrates. We used the same size and distribution of food patches as in our simulations (*Figure 3A*). Food was distributed inside, with agar outside patches (*Figure 3A* and Materials and methods). We tested sucrose and yeast at the same concentration as in the homogenous substrate. We also performed experiments using apple juice as a nutrient, motivated by the fact that it is ecologically relevant and that, unlike sucrose, the fructose contained in apple juice is volatile, which makes it detectable by smell and not only by taste. Groups of five larvae of the same type (rovers or sitters) were placed inside each patch (total of two) at the beginning of the recordings (total of ten larvae of the same type *per replicate*, repeated in three independent experiments).

We tracked the trajectories with the same methods used in the homogeneous environment (*Figure 3B* and *Figure 3—figure supplement 1A*). Then, we performed the analysis separately for the two different regions: inside and outside the patches, and quantified features of the larval exploratory behavior. Inside yeast and apple juice patches, larvae crawled significantly slower than outside them (*Figure 3C*). In yeast patches, both rovers and sitters executed fewer turns inside than outside (*Figure 3D*). All larvae made significantly more pauses inside the food patches than outside (*Figure 3E*). We also observed that the handedness score of the larvae is less broad than in the homogeneous substrates (*Figure 3F*), which may be caused by reorientations that are triggered to prevent the larva from exiting the food patch. As expected from the phenotype, sitter larvae crawled a shorter distance in the first 5 min of the recording in the yeast but also the sucrose substrates (*Figure 3G*). In general, sitter larvae had slower crawling speeds and executed fewer turns in the patchy environments than rovers (*Figure 3—figure supplement 1A–C*). We also noticed that sitters paused more inside patches than rovers (*Figure 3—figure supplement 1D*). Outside yeast and apple juice patches, the crawling speed increased but did not return to levels similar to the agar-only condition, suggesting that the behavior of larvae that exit the patch is influenced by the recent food experience or that larvae might still be sensing the food (*Figure 3—figure supplement 1E*). In line with this, in yeast the number of turns outside the patch was higher than inside the patch.

Our model predicted that fraction of time spent inside patches should vary according to the substrate: larvae should remain longer inside yeast patches than inside sucrose patches (*Figure 2E*). In particular, simulated sitter larvae stayed longer than simulated rovers inside yeast patches. In the experiments, the same trend was observed: for both rovers and sitters the fraction of time spent inside patches was higher in the yeast compared to both sucrose and apple juice patches (*Figure 3H*).

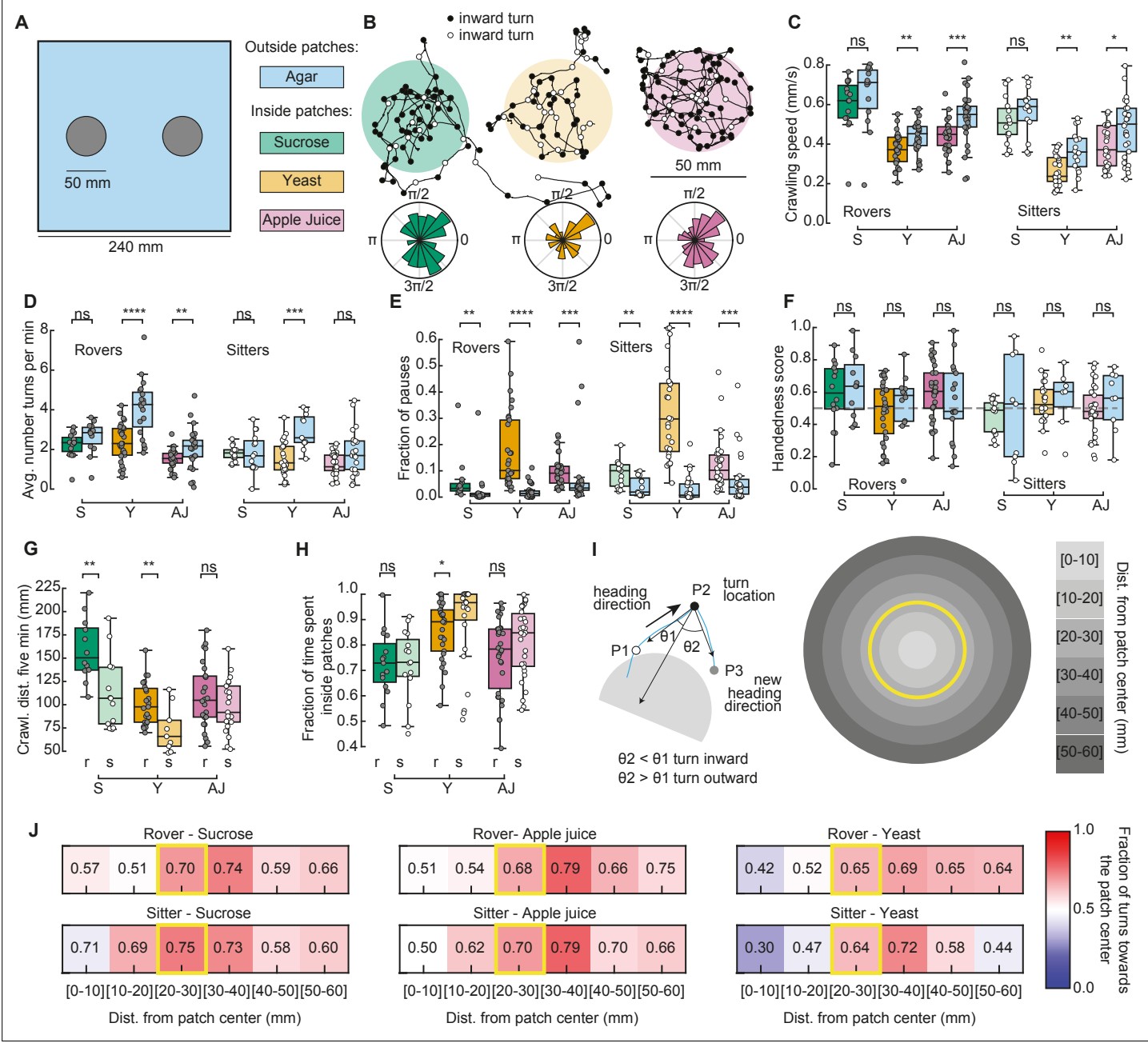

**Figure 3.** Larval exploratory behavior in patchy substrates. (**A**) Experimental setup: Five larvae of the same phenotype were placed on top of each food patch (two patches, total: 10 larvae per experiment). Three types of food patches were tested: sucrose (green), yeast (orange), and apple juice solution (magenta). Agar was uniformly spread in the arena outside the food patches. (**B**) Sample trajectories of sitter larvae in the three patch substrates with inward (outward) turns marked in black (white) circles. Distribution of turning directions is shown on the bottom of each trajectory. (**C**) Larval crawling speeds of rovers and sitters measured inside (colored bars) and outside (blue bars) food patches: sucrose (S, green), yeast (Y, orange), and apple juice (AJ, magenta). Horizontal line indicates median, the box is drawn between the 25th and 75th percentiles, whiskers extend above and below the box to the most extreme data points that are within a distance to the box equal to 1.5 times the interquartile range and points indicate all data points. (**D**) Average number of turns executed per minute. (**E**) Fraction of time in which larvae did not move (pauses). (**F**) Handedness score. (**G**) Total distance crawled by rover (r; darker colors) and sitter (s; lighter colors) larvae in the first 5 min of the recording. (**H**) Fraction of time spent inside patches of rovers (r) and sitters (s). (**I**) Left: Identification of turning angle as inwards ($\theta_2 < \theta_1$, black) or outwards ($\theta_2 > \theta_1$, white). Right: Circular regions with fixed distances relative to the patch center. The yellow line represents the patch border. (**J**) Relative fraction of inward turns calculated as a function of the distance from the patch center. The distance bin that includes the patch radius is highlighted in yellow. Left: Sucrose, middle: apple juice, right: yeast patches. Top: Rovers, bottom: sitters. Mann–Whitney–Wilcoxon test two-sided was performed since the data are not normally distributed. ns: 0.05 < p < 1, *0.01 < p

*Figure 3 continued on next page*

*Figure 3 continued*

< 0.05, **0.001 < p < 0.01, ***0.0001 < p < 0.001, ****p < 0.0001. The total number of larvae tested is detailed in *Table 1*. Statistical power and Cohen's size effect of non-significant comparisons are included in *Table 4*.

The online version of this article includes the following figure supplement(s) for figure 3:

**Figure supplement 1.** Comparison between rover and sitter behavior in two patch substrates.

**Figure supplement 2.** Larval exploratory behavior in no-food patchy substrates.

Sitter larvae stayed significantly longer inside yeast patches than rovers (*Figure 3H*). Nevertheless, the percentage of time the larvae spent inside patches in the experiments was very different from our model predictions. Rover (sitter) larvae remained on average 72.6% (72.3%) of the experiment inside sucrose, 85.7% (90.0%) inside yeast, and 75.6% (81.3%) inside apple juice patches. Those values were much higher in the experiments than what we predicted with our simulations, and suggest that larvae might employ other mechanisms in addition to slower crawling and more frequent pauses to remain inside the food.

To gain more insight into the strategies used by larvae to increase the time spent inside the food patches, we studied the distribution of turns in the food–no food interface. First, we labeled each turn as inwards or outwards depending on whether they were oriented toward or away from the patch center (*Tao et al., 2020*; *Figure 3I*, left). We observed that inward turns occur more often than outward turns at the border of the patch for the three substrates (*Figure 3B*, inward turns are shown in black). To control for possible mechanosensory effects due to the border edges, we prepared new arenas with patches that contained no nutrients, either using the same agar that composed the rest of the arena, or using ultrasound gel (Methods). Larvae in the agar–agar or the agar–gel border did not show any changes in their preference to turn toward the patch center, confirming that the behavioral change observed in response to food is specific (*Figure 3—figure supplement 2*).

We then studied the fraction of turns toward the patch center as a function of the distance to the patch center (*Figure 3I*, right). For the three types of substrates, the bias to turn inwards was clearly manifested when the larvae experienced the patch border (patch radius: 25 mm, distance bin: 20–30 mm) (*Figure 3J*). The bias persisted when the larva exited the patch (distance bins: 30–40, 40–50, 50–60 mm). We did not consider further distance bins in our analysis because most larvae did not reach those locations in our experiments.

Therefore, our model predictions do not seem to be well supported by experiments with patchy substrates. In particular, we conclude that when larvae reach the food–no food interface their turning behavior changes. This is accomplished by turning toward the patch center while maintaining the handedness (*Figure 3J* and *Figure 3—figure supplement 1F*) and represents an important mechanism to remain inside the food.

## Anosmic larvae also select turns toward the patch center when reaching the food–no food border, but not on the yeast

It is well known that *Drosophila* larvae can efficiently navigate toward or away an odor source using chemotaxis (*Louis et al., 2008*; *Gomez-Marin et al., 2011*; *Schulze et al., 2015*). Chemosensory information from gustatory and olfactory receptors is combined to allow larvae to locate food sources in the environment (*Vosshall and Stocker, 2007*). We next wondered how much of the tendency to turn toward the patch center once outside the patch could be attributed to processing olfactory cues.

Thus, we repeated the patchy experiments with mutant anosmic larvae, where Orco, the obligatory co-receptor for all olfactory neurons, apart the $CO_2$ sensing ones, is mutated (*Vosshall and Stocker, 2007*) and tested if they show the same distant-dependent bias when exploring the patchy substrate.

Anosmic larvae extensively explored the patchy substrate (*Figure 4A*). In general, they exhibited a small difference in crawling speeds when comparing their behavior inside vs. outside of food patches (*Figure 4B*). Curiously, this difference in speeds was non-significant inside vs. outside yeast patches. We also found that the fraction of pauses of anosmic larvae in yeast patches was smaller than that of rovers and sitters (*Figures 3G and 4D*). This suggests that yeast patches are not attractive to anosmic larvae, in agreement with the lower fraction of time spent inside yeast patches relative to sucrose and apple juice patches (*Figure 4F*).

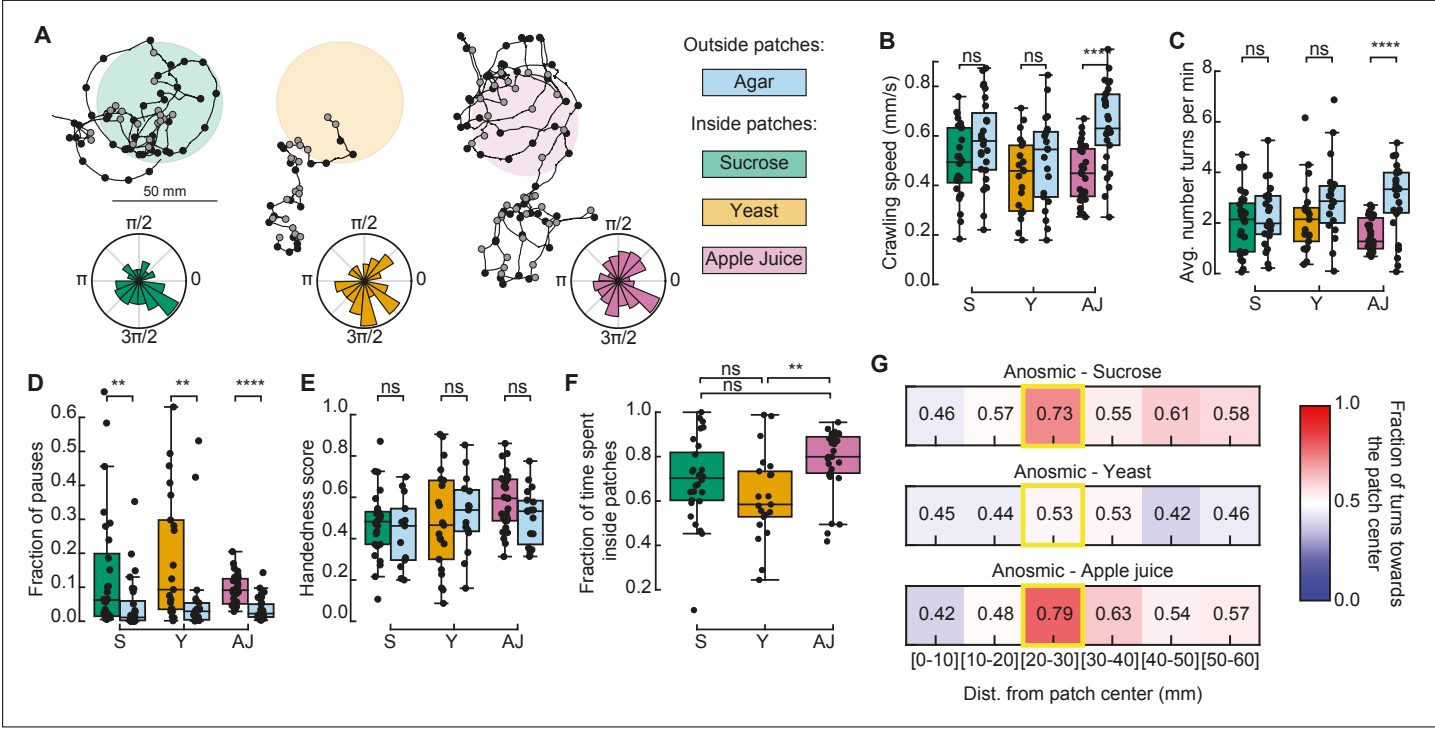

**Figure 4.** Exploratory behavior of anosmic larvae in patchy environments. (**A**) Sample trajectories of anosmic larvae in the three patch substrates with inward (outward) turns marked in red (gray) circles. Distribution of turning directions is shown on the bottom of each trajectory. (**B**) Crawling speeds of anosmic larvae measured inside (colorful bars) and outside (blue bars) food patches: sucrose (S, green), yeast (Y, orange), and apple juice (AJ, magenta). Horizontal line indicates median, the box is drawn between the 25th and 75th percentiles, whiskers extend above and below the box to the most extreme data points that are within a distance to the box equal to 1.5 times the interquartile range and points indicate all data points. (**C**) Average number of turns per minute inside and outside patches. (**D**) Fraction of pauses inside and outside patches. (**E**) Handedness score of anosmic larvae inside and outside patches. (**F**) Fraction of time spent inside patches for different types of food. (**G**) Relative fraction of inward turns calculated as a function of the distance from the patch center, top: sucrose, middle: yeast, bottom: apple juice. The distance bin that includes the patch radius is highlighted in yellow. Mann–Whitney–Wilcoxon test two-sided was performed since the data are not normally distributed. ns: $0.05 < p < 1$, **$0.001 < p < 0.01$, ***$0.0001 < p < 0.001$, ****$p < 0.00001$. The number of larvae tested is detailed in **Table 1**. Statistical power and Cohen's size effect of non-significant comparisons are included in **Table 4**.

The online version of this article includes the following figure supplement(s) for figure 4:

**Figure supplement 1.** Analysis of diffusion of nutrients on behavior.

Next, we investigated if anosmic larvae can bias their turns at the patch border interface without navigating odorant cues. Turns in the trajectory were labeled as inwards or outwards (as in **Figure 3I**) and the fraction of turns toward the patch center was analyzed as a function of the distance away from the patch center.

In sucrose and apple juice substrates, anosmic larvae consistently increased the fraction of inward turns near the patch border (20–30 mm; **Figure 4G**). This was not the case in the yeast patches, where no bias was detected at the patch border.

In sum, we found that anosmic larvae, apart from on yeast, trigger turns toward the patch center at the food–no food interface, suggesting that olfaction is not the only mechanism responsible for the turning bias that increases the fraction of time larvae spend inside patches.

Taste very likely influences the probability that larvae remain in the patches. To control for the diffusion of nutrients (sucrose and apple juice) at the edge of a patch, we evaluated the maximum distance at which an increased fraction of turns toward the center was significantly different when compared to the yeast non-responsive anosmic control. At a distance greater than 0.5 cm from the edge, anosmic larvae on sucrose, apple juice, and yeast were indistinguishable, suggesting that diffusion has a limited impact on behavior (**Figure 4—figure supplement 1A**).

Finally, to control for possible effects of diffusion over time, we compared the fraction of turns toward the center in the first and second half of the experiment. For most distance and nutrients, the two distributions were not significantly different (**Figure 4—figure supplement 1B**).

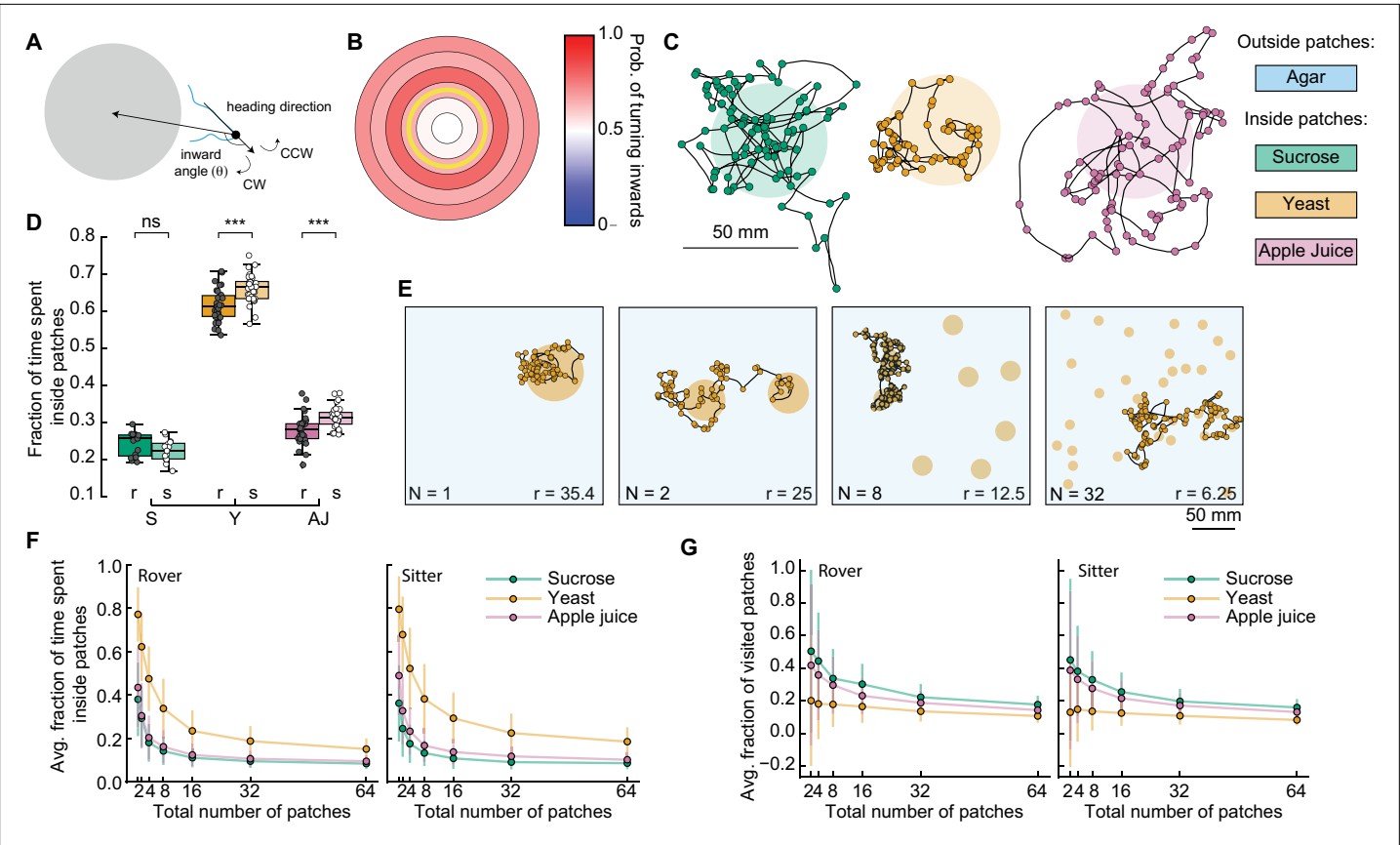

**Figure 5.** Interplay between food quality and patches distribution. (**A**) Schematic showing inward turn (clockwise, CW) being selected by the simulated larva. By selecting inward turns, the trajectory approaches the patch center. (**B**) Spatial-dependent probability of turning toward the patch center. Each region is a concentric circle with a fixed probability of drawing inward turns (see *Figure 3I*, right). The yellow line shows the patch border. (**C**) Sample simulated trajectories for a sitter larva with biased inward turns: sucrose patch (green), yeast patch (orange), and apple juice patch (magenta). (**D**) Fraction of time spent inside patches of rovers (r; darker colors) and sitters (s; lighter colors) in the different substrates: sucrose (S, green), yeast (Y, orange), and apple juice (AJ, magenta). Each point is 30 simulation runs of one larva (total: 30 larvae simulated per substrate). Horizontal line indicates median, the box is drawn between the 25th and 75th percentiles, whiskers extend above and below the box to the most extreme data points that are within a distance to the box equal to 1.5 times the interquartile range and points indicate all data points. (**E**) Sample trajectories of sitter larvae in environments with varying number of randomly located patches, with a fixed total area of yeast substrate being distributed (Np = 1, 2, 8, 32 from left to right). (**F**) Average fraction of time spent inside patches of distinct substrates (S: sucrose, green; Y: yeast, orange, and A: apple juice, magenta) for rovers (left) and sitters (right) as a function of the number of patches. Each point is the average of 30 larvae (30 simulation runs each). Bars show the standard deviation. (**G**) Same as (F) but for the average fraction of visited patches. Mann–Whitney–Wilcoxon paired test was performed since the data are not normally distributed. ns: 0.05 < p < 1, ***0.0001 < p < 0.001.

The online version of this article includes the following figure supplement(s) for figure 5:

**Figure supplement 1.** Fraction of time spent inside patches with the model that includes turn bias.

**Figure supplement 2.** Simulations with varying number of patches.

## To remain inside of the food patch larvae combine turning bias with other strategies

To understand the impact of the turning bias on the percentage of time that larvae spend inside patches, we included a distance-dependent probability of turning toward the patch center in our model (*Figure 5A*). After drawing a turning angle from the probability distribution, the turn was implemented toward the patch center with probability $P_{bias}$ that depends on the distance between the current position and the center of the closest patch (*Figure 5B*). For each simulated substrate, larva type, and relative distance, $P_{bias}$ corresponds to the fraction of turns toward the patch center quantified in our experiments (*Figures 3J and 4G*).

We observed that the simulated trajectories with this distance-dependent turning bias resemble the experimental ones much more (*Figure 5C*), with larvae often returning to a patch when leaving its border. Indeed, larvae spent three times longer inside a patch in the new simulations compared to the model without biased orientations (*Figure 5D* and *Figure 5—figure supplement 1A*): now rover (sitter) larvae remain on average 31.1% (28.9%) of the simulation inside sucrose patches and 63.8 (68.4%) of the simulation inside the yeast patches. Simulated anosmic larvae also showed a gain in the ratio of time inside patches (*Figure 5—figure supplement 1B, C*). Therefore, biased orientations at the patch border are an important mechanism employed by larvae to return to a food source when they detect a change in the substrate quality. This can be achieved without olfactory orientation cues, since anosmic animals can also perform biased turns (*Figure 4G*). However, the ratio of time that simulated larvae remain inside patches was still smaller than that measured in the experiments (*Figures 3H and 4F*). We reason that other mechanisms, such as working or short-term memory (*Louis et al., 2008*; *Schleyer et al., 2015*), or other sensory modalities at the vicinity of the border of the patch (see discussion) can contribute to increasing the time inside the food.

## Our model reveals the interplay between food quality and patches fragmentation

We next used our model to investigate how a further fragmentation of the food patches affects the ability of larvae to stay in patches where they can feed. To test we fixed the total area of food S and varied the number of patches choosing the center coordinates for each patch randomly (*Figure 5E*). We tested seven levels of fragmentation from 1 to 64 patches and to compensate for different patch radii, we adjusted the distance-dependent probability to turn inwards of each larva (*Figures 3J and 4G*, see Materials and methods). We modeled the three types of food tested thus far, for rover, sitter, and anosmic larvae. In total, this would represent 1575 hr of experiment, highlighting the advantage of the model.

First, we quantified the average fraction of the time spent inside patches relative to the whole simulation for the different food substrates as a function of the number of patches (*Figure 5F* and *Figure 5—figure supplement 2A, B*). As expected, both rovers and sitters spent less time inside a patch as the number of patches increases (and thus the patches radius decreases) (*Figure 5F*). Larvae spent longer inside patches in more nutritious environments, for example yeast, irrespective of the number of available patches. Interestingly, despite the small differences we previously quantified, our results showed that sitter larvae consistently spent more time inside yeast patches than rovers for each number of patches (*Figure 5—figure supplement 2D*). This was not observed in the sucrose or apple juice patches. Anosmic animals also spent less time inside patches when the number of patches increases, but the dependence on the quality of food was much less pronounced (*Figure 5—figure supplement 2B*).

Next, we investigated the effect of different food substrates on the number of patches larvae explore to understand how fractioning environment would affect exploitation, which is key for survival. We quantified the fraction of new patches a larva visits during the simulation (discounting the source patch, since all the simulations start with the larva inside one patch) (*Figure 5G*). Rovers and sitters explored more patches in the less nutritious substrate (sucrose), with a slightly higher fraction of visited patches for rovers in the sucrose and yeast patches (*Figure 5G*; *Figure 5—figure supplement 1E*). Anosmic larvae showed a weaker effect of the substrate on the fraction of patches visited (*Figure 5—figure supplement 1C*).

Our model predicts a trade-off between the quality of the food and the fraction of patches visited: when exploring a substrate with low-quality (high-quality) food, the larvae are more (less) likely to leave and more (fewer) patches are visited.

## Larvae experience a trade-off between food consumption and exploration

To confirm that larvae adapt their behavior as modeled in response to different quality and fragmentation of food, we compared the behavior of larvae in two and eight patches. We conducted new experiments in arenas with eight patches of sucrose and yeast with rover and sitter larvae. Three sets of random positions of patches were used for each replicate (*Figure 6B*). Each larva (total of eight)

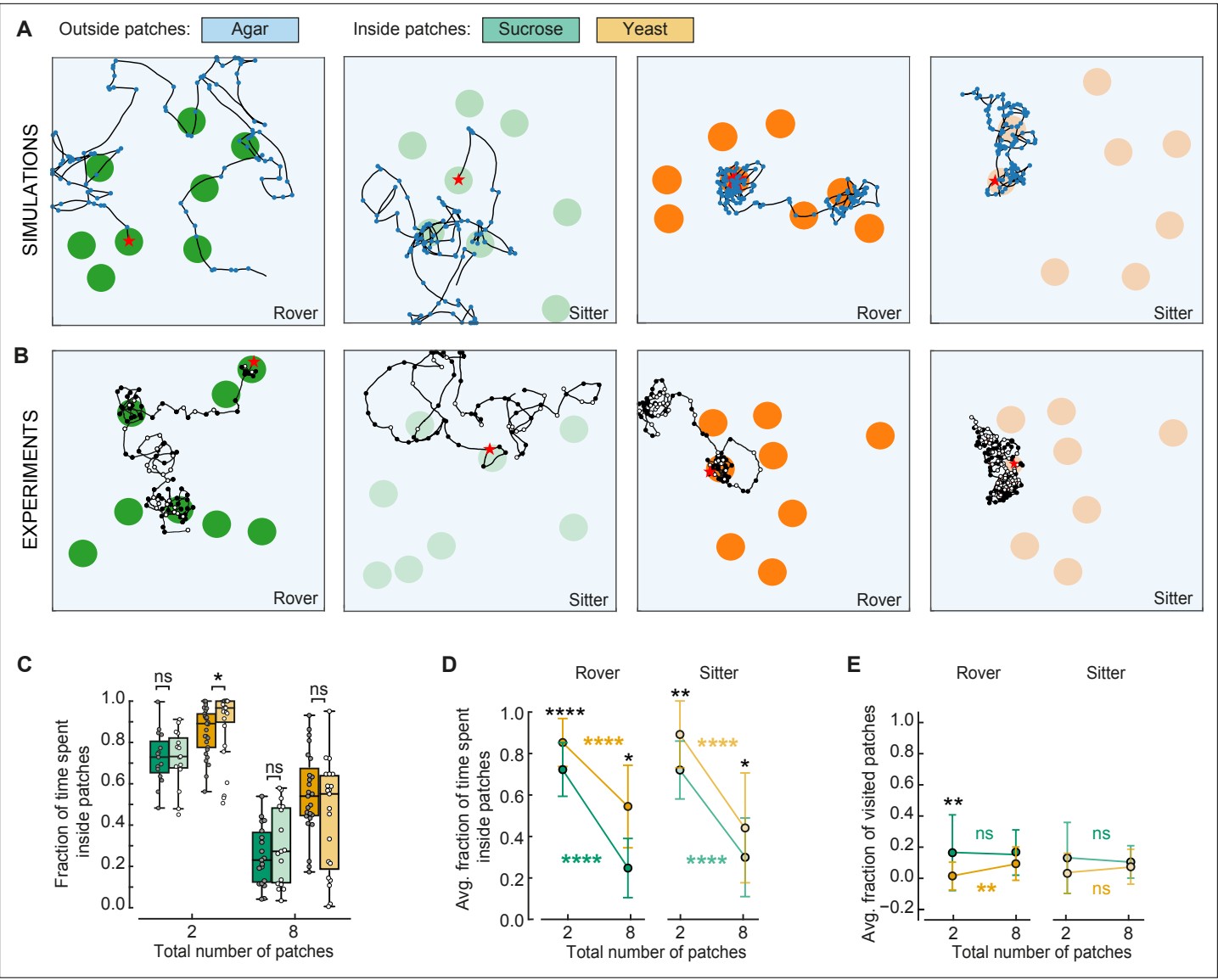

**Figure 6.** Behavioral response to changes in food quality and fragmentation. (**A**) Sample simulated trajectories for sitter and rover larvae exploring in eight patches: sucrose patch (green) and yeast patch (orange). (**B**) Sample experimental trajectories of rover and sitter larvae in an arena with eight patches of food. Three random distributions (exp1; exp2; exp3) were used for each type of food: sucrose patches (green) and yeast patches (orange). (**C**) Fraction of time spent inside patches of rovers (darker colors) and sitters (lighter colors) on sucrose (green) and yeast (orange). Horizontal line indicates median, the box is drawn between the 25th and 75th percentiles, whiskers extend above and below the box to the most extreme data points that are within a distance to the box equal to 1.5 times the interquartile range and points indicate all data points. (**D**) Average fraction of time spent inside patches of distinct substrates (sucrose, green; yeast, orange) for rovers and sitters as a function of the number of patches. Data represent mean ± standard deviation. (**E**) Same as (D) but for the average fraction of visited patches. analysis of variance (ANOVA) test was performed for (C) and (D) (normally distributed) and Mann–Whitney–Wilcoxon paired for (E) (non-normally distributed). ns: $0.05 < p < 1$, *$0.01 < p < 0.05$, **$0.001 < p < 0.01$, ****$0.00001 < p < 0.0001$. The number of larvae tested is detailed in **Table 1**. Statistical power and Cohen's size effect of non-significant comparisons are included in **Table 4**.

was placed inside a different patch and left to crawl for 50 min. The data were compared to the experiments with two patches (**Figure 3**).

A first comparison of the trajectories of simulated and experimental larvae exploring in an environment with eight patches shows great similarity (**Figure 6A, B**). As predicted by the model, both rovers and sitter spent half of the time inside patches when the area of food was divided in eight compared to two patches (**Figures 5F and 6C, D**). Furthermore, the larvae stayed longer on the yeast patches

compared to the sucrose ones (*Figure 6C, D*), supporting the prediction of the model that larvae will spend less time in less nutritious patches irrespective of the number of available patches.

We then analyzed the effect of food quality on the proportion of patches visited by the larvae. There were no significant differences comparing the larvae in yeast and sucrose apart for rover in yeast for two patches. In this case, the model had predicted a difference between yeast and sucrose that is not present experimentally, probably because the larvae spend more time on the patches than what the model predicted via other mechanisms. However, it is clear that larvae spent more time looking for new patches (outside patches, *Figure 6C, D*) when the quality of food was lower (in sucrose) compared to higher quality (yeast), but they did not reach more patches in our experimental timeline. It is possible that having left a source of poor food, the larvae were more interested in exploring in search of food of better quality.

Finally, we were particularly interested in testing the prediction that larvae would reach a steady state in the proportion of patches visited as the food would become more fragmented. This was supported by the experiments with two and eight patches despite our suspicion that.

Overall, the experiments show how larvae tune the elements of the navigation routine to generate a foraging behavior that adapts to the quality and spatial distribution of food resources.

## Discussion

Foraging behavior is a complex process influenced by many internal factors (locomotion style, sensory perception, cognitive capacity, age) and external variables (spatiotemporal distribution of resources, presence of predators, social interactions with co-specifics). Here, we focused on the detailed characterization of foraging in a single model organism, the fruit fly *Drosophila* larva, using extensive experiments and modeling. This allowed us to study the role of both internal and external factors on foraging: (1) genetics (rovers, sitters, and later *orco* null anosmic animals), (2) food quality (agar, yeast, sucrose, and apple juice), and (3) food spatial distribution (homogeneous and heterogeneous environments).

We systematically investigated larval exploratory behavior first in experimental arenas with homogeneously distributed food. Larval crawling speed, turning frequency and fraction of pausing events adapted according to the quality of the food substrate (*Figure 1C–E*). The quality of the food had a strong impact on the distance traveled by the larvae. In yeast, larvae moved less and their speed and turn frequency were decreased. They also made more pauses, with the majority remaining stationary, except for internal gut movements (*Video 1*), which suggested that they were digesting the yeast. The pauses were rarely observed in sucrose, which is metabolized more quickly than yeast, even when mixed with agar (*Figure 1E*).

We observed that larval trajectories often had a circular shape, revealing an individual preference for a given turning direction in the absence of direction cues, which we quantified as the larval handedness (*Figure 1B, F*). The population variability in the handedness has been quantified in adult flies (*Buchanan et al., 2015*), but to our knowledge not until now at the larval stage. In adult walking flies, individual preferences of turning left or right in maze tests have been shown to persist across days (*Buchanan et al., 2015*) and recently have been linked to anatomical differences in the synaptic distribution of bottleneck neurons downstream of the central complex (*Skutt-Kakaria et al., 2019*). It is therefore possible that, as found in adults, larval individual differences in neuronal connections could define handedness. It would be interesting to understand the evolutionary advantage of handedness, if there is one, and to relate it to the 'hard-wired' circuitry controlling Lévy search behavior (*Sims et al., 2019*).

It is expected that animals change their foraging behavior depending on the quality and spatial distribution of food, with more localized exploitation of resources where they are abundant and a more exploratory behavior when resources become scarce (*Humphries et al., 2010*). We tested this in a phenomenological model of larval foraging behavior in patchy substrates (*Humphries et al., 2010*; *Figure 2*). We reasoned that crawling speed, turning frequency and fraction of pauses are the behavioral elements that adapt when the larva crosses the food–no food interface at the patch boundary. To quantify the food exploitation, we measured the fraction of the time each larva spent inside the patches. We found that decreasing the speed and turning frequency and increasing the fraction of pauses is not sufficient to explain why larvae remain inside the food for longer periods.

In experiments with patchy substrates, we found that larvae spend a longer time inside food patches than predicted with our model (*Figure 3H*). The lack of agreement between the experiments and our model was not surprising, since the latter does not include additional mechanisms that could guide the larva back to the patch when it leaves it, such as chemotaxis (*Louis et al., 2008*; *Gomez-Marin et al., 2011*). Since in chemotaxis larvae redirect their turns toward a source of odor, we classified each turn in their recorded trajectory as toward or away from the patch center. We observed that the fraction of inward turns is very high around the patch border (*Figure 3J*). To test whether larvae could redirect their turns toward the food when exiting it using olfactory cues, we repeated the experiments with anosmic mutants. Surprisingly, in sucrose and apple juice substrate anosmic larvae bias their turns toward the patch center when in the neighborhood of the patch border (*Figure 4G*). Therefore, this reorientation at the border does not seem to rely solely on olfaction. When exiting the food patch, larvae sense the lack of taste and it is possible that the turn bias changes as a result of temporal integration of the recent sensory-motor experience allowing them to return to the patch, as observed when navigating in an olfactory or light intensity gradient. Also, the patches of sucrose and apple juice were in direct contact with the surrounding agar arena. This has the advantage of generating a smooth transition in the substrate (*Figure 3—figure supplement 1E, F*), but it also allows diffusion at the interface which the larvae can sense as they crawl away from the food (*Lebrun and Junter, 1993*). In anosmic larvae, the fraction of turns toward the center for sucrose and apple juice patches was only higher compared to the one for the yeast patch (where there was no food effect) within the first half centimeter outside the patch, suggesting that the impact of diffusion could be significant only in that region (*Figure 4—figure supplement 1*).

An experiment using the gustatory sweet sensor *Gr43a* mutant on sucrose, which is not volatile and does not produce smell, could help discerning the contribution of taste at the border of the patch (*Fujishiro et al., 1984*; *Marella et al., 2006*; *Miyamoto et al., 2013*; *Wang et al., 2004*; *Mishra et al., 2013*). For yeast, the lack of smell completely changed the response of the larvae, which did not show differences inside and outside the patch for most foraging parameters (*Figure 4B, C, E, G*). In this instance, taste was not sufficient to retain larvae inside the yeast patch (compare *Figure 3H* with *Figure 4F*) even though several gustatory receptors have been shown to be activated by yeast metabolites (*Wisotsky et al., 2011*; *Ganguly et al., 2017*; *Croset et al., 2016*).

Another sensory modality that could have influenced the larval behavior at the food–no food interface, is mechanosensation. We excluded the possible role of the border of the patches performing experiments in patches without food (*Figure 3—figure supplement 2*). However, when larvae are crawling, they leave a print of their denticle attachment on the agar, that could inform them about their previous location and help returning to the food. Overall, the differences in behavior of larvae exposed to different foods, revealed the complexity of the sensory-motor processing involved in foraging.

One of the strengths of our phenomenological model is that it incorporates a modular organization of foraging that could reflect how the crawl and turn modules are controlled. First, we modeled a stochastic search where no information regarding food is available outside of the current location, because food is absent or because the larvae cannot sense it. This corresponds to an autonomous search behavior implemented by circuits located in the ventral nerve cord without input from the brain (*Berni et al., 2012*; *Sims et al., 2019*). Second, we incorporated a goal-directed navigation that allows larvae to return to the food. Our phenomenological model includes a distance-dependent probability to turn inwards that mimics the effect of chemotaxis (when present), as much as any other possible mechanism that contributes to the turning probability. As a consequence, we observed that simulated larvae, even when the resources are fractioned in eight patches, could stay inside the food patch for longer periods, in line with experimental observations (*Figures 5 and 6*). The model could be improved by setting the turning properties outside the patch to match as closely as possible experimental observations. To this end, we could consider studies of larvae crawling in different attractive gradients, where the changes in turning probability and angle, including weathervaning, have been investigated in relation to precise spatiotemporal information of odorants (*Louis et al., 2008*; *Gomez-Marin et al., 2011*; *Davies et al., 2015*). It would also be helpful to have information about other attractive gradients, like taste, to know if a common set of mechanisms is used regardless of the sensory modality. Using this information, our model could be used to investigate how crawling speed and turning properties are controlled via descending pathways from the brain (*Tastekin et al.,*

*2018*; *Jovanic et al., 2019*). Finally, in the presence of nutrients, our model adjusts movements to stay on the food patch. The concerted decrease in turning rate and crawling speed and increase in the number of pauses, suggests that a neuromodulatory depression of movement (*Marder, 2012*; *Lin et al., 2019*) could be relevant in this phase. It would be interesting to investigate more generally how neuromodulators influence the decision to remain or explore new food resources in relation to the resources available and the larval motivational state.

Overall, we found both in our experiments and modeling that larvae spend less time exploiting patches of less nutritious food (e.g., sucrose). What could be the effect of this when several patches are available in the substrate? Our model results predict that larvae would spend more time exploring and more patches would be visited when food quality is lower (*Figure 5G*). In natural environments, this would enhance the chances that larvae will eventually find a better food source in the surroundings. Our experiments show a slightly different picture, where larvae indeed explore for a longer period when on less nutritious food but the number of patches they find is not increased compared to when they are on a more nutritious food (*Figure 6C, D*). It is possible that having left a poor food source, the larvae are more likely to continue looking for a more nutritious one, in the short term, instead of visiting and exploiting a new poor patch. Therefore, the internal state of the animal is probably playing an important role in the decision of choosing a new patch of food to exploit (*Ringo, 2018*; *Branch and Shen, 2017*).

The differences we found in the foraging behavior of rovers and sitters are not as drastic as previously reported, where the length of the path of rovers was roughly twice that of sitters when crawling in a yeast paste for 5 min (*Sokolowski, 2001*). In the homogeneous agar, sucrose, and yeast substrates, we did not observe significant differences in the path length of rovers and sitters (*Figure 1—figure supplement 1*). This was expected for the no-food condition (agar substrate; *Kaun et al., 2007*; *Yang et al., 2000*), but not in the presence of yeast (*Sokolowski, 2001*). This could be attributed to differences in the food preparation protocol: we applied a thin layer of yeast on top of the agar surface instead of thick yeast suspension as in *Sokolowski, 1980* to allow recording from underneath the food (*Risse et al., 2013*). Also, our experiments were conducted in the dark, which might influence behavior (*Sokolowski, 1980*).

Interestingly, when the food is constrained inside patches, as done in the classical work studying the foraging polymorphism, we observed significantly shorter crawling paths of sitters in sucrose and yeast patches (*Figure 3G*). Sitters' crawling speed was also slower and they perfomed fewer turns *per* minute and more pauses (*Figure 3—figure supplement 1*). It is possible that the presence of a patch border plays a significant role for the foraging polymorphism phenotypic expression.

In summary, we have identified a set of behavioral elements – the crawling speed, frequency and biasing of turns, and fraction of pauses – that adapt when larvae explore environments with a patchy distribution of food sources. This adaptation leads to an efficient substrate exploration, as larvae either increase the time inside nutritious food patches or continue exploring the substrate depending on the local quality of food.

## Materials and methods

### Animals

Rover and sitter flies were a gift of Marla Sokolowski (University of Toronto) and Orco[2] from Bloomington stock center (stock 23130). Flies were allowed to lay eggs for 1 day in standard corn meal food, which consists of 420 g of cornmeal; 450 g of dextrose; 90 g of yeast; 42 g of agar; 140 ml of 10% Nipagin in 95% EtOH; 22 ml of propionic acid, and 6.4 l of water. Larvae that were 72 hr old were collected for the experiment.

### Larva tracking

We recorded movies of larval exploratory behavior in arenas with minimal external stimuli – the recordings were made in the dark with a constant temperature of 25°C. Each trial lasted 50 min and the larvae were simultaneously tracked in a $240 \times 240$ mm$^2$ arena with a 2-mm thick layer of 0.4% agar-based coating (see the protocol of substrate preparation below).

At each trial, 10 young third-instar larvae (72–80 hr since egg laying) of approximately the same size were washed to remove traces of food and allowed to crawl freely for 5 min on a clean 0.4%

**Table 1.** Number of larvae per recording.

| Substrate | Number of trials | Average number of larvae per trial | Total larvae |
|---|---|---|---|
| Agar homogeneous | 3 rovers<br>3 sitters | 10 rovers<br>10 sitters | 30 rovers<br>29 sitters |
| Sucrose homogeneous | 3 rovers<br>3 sitters | 10 rovers<br>10 sitters | 30 rovers<br>30 sitters |
| Yeast homogeneous | 3 rovers<br>3 sitters | 10 rovers<br>10 sitters | 29 rovers<br>30 sitters |
| Agar 2 patches | 3 sitters | 9 sitters | 28 sitters |
| Gel 2 patches | 3 sitters | 11 sitters | 33 sitters |
| Sucrose 2 patches | 2 rovers<br>2 sitters<br>3 anosmic | 8 rovers<br>8 sitters<br>10 anosmic | 15 rovers<br>15 sitters<br>28 anosmic |
| Sucrose 8 patches | 3 rovers<br>3 sitters | 8 rovers<br>8 sitters | 24 rovers<br>19 sitters |
| Yeast 2 patches | 3 rovers<br>3 sitters<br>3 anosmic | 10 rovers<br>10 sitters<br>10 anosmic | 30 rovers<br>27 sitters<br>21 anosmic |
| Yeast 8 patches | 3 rovers<br>3 sitters | 8 rovers<br>8 sitters | 25 rovers<br>21 sitters |
| Apple juice 2 patches | 3 rovers<br>3 sitters<br>3 anosmic | 10 rovers<br>10 sitters<br>10 anosmic | 30 rovers<br>30 sitters<br>29 anosmic |

agar coated plate before being transferred to the arena (*Table 1*). We used a Frustrated Total Internal Reflection (FTIR)-based imaging method to record the larval exploratory behavior (*Risse et al., 2013*). Movies (duration 50 min) were recorded with a Basler acA2040-180km CMOS camera at 2048 × 2048 px$^2$ resolution, using Pylon and StreamPix software, mounted with a 16-mm KOWA IJM3sHC. SW VIS-NIR Lens and 825-nm high-performance longpass filter (Schneider, IF-093). We recorded the movies at 2 frames per second to obtain forward movement displacements and actual pause turns that are recorded accurately rather than to include 'flickering' movements associated with peristaltic movements.

## Substrate preparation

The following food substrates were prepared for our experiments, and stored refrigerated for up to 1 day:

1. Agar substrate: 0.8 g of agar was melted in 200 ml of distilled water;
2. Sucrose substrate: 0.8 g of agar with 3.42 g of sucrose was dissolved in 200 ml of distilled water;
3. Apple juice substrate: 0.8 g of agar with 0.342 g of sucrose and 5 ml apple juice (Del Monte Quality Pure Apple Juice from Concentrate) was dissolved in 195 ml of distilled water;
4. Yeast substrate: 0.8 g of agar was melted in 200 ml of distilled water with a layer of 5 ml of 20% yeast in water on top.

In the case of agar and sucrose homogeneous substrates, the solution was homogeneously spread on top of the acrylic arena and we waited for it to reach room temperature before transferring the larvae to the arena. Yeast homogeneous arenas were obtained by spreading 5 ml of 20% yeast in water with a soft metallic disk. For sucrose or apple juice patchy arenas, first, the agar solution was homogeneously spread in the acrylic arena. When the solution cooled down, two holes in the agar were made at fixed positions (*Figure 3A*) using circular-shaped Petri dishes with a 25-mm radius. We carefully removed the agar inside the holes and transferred the food solutions to the holes with the same thickness as the agar around them. Control two agar patches were filled with 04% agar alone. For each one of the two yeast patches, 100 µl of yeast solution was placed on a 25-mm-radius metal disc and printed on the agar. For gel 2 patches control we stamped a drop of 150–200 mg of ultrasound

gel for TENS machine (Boots ingredients: purified water, glycerin, propylen glycol, hydroxyethylcellulose, sodium citrate, citric acid, domiphen bromide). The viscosity of the gel is not identical to the one of yeast, but it informs us about the transition from viscous and smooth (gel-yeast) to agar. For eight patches, we chose three distributions randomly generated in the modeling experiment (*Figure 6A*, lower panel). Using a 12.5-mm-radius disc we printed the patches with 25 µl of yeast solution. For sucrose 8 holes were made using a cylinder and then filled with food solution. One larva was placed in each patch, meaning that each larva was exposed to a different distribution of the resources. The experiments were repeated three times.

## Descriptive statistics of larval trajectory

The data (*x,y* coordinates of individual larvae) were extracted from the behavioral movies using the FIM track free software (*Risse et al., 2017*). We used a Kalman filter to the (*x,y*) coordinates of each larva (code will be available at github after the paper is accepted). The position of each larva in video frame *j* is represented as the vector:

$$\vec{R}\left(t_j\right) = \left(x\left(t_j\right), y\left(t_j\right)\right), j = 1, 2, \ldots, N$$

where $x\left(t_j\right)$ and $y\left(t_j\right)$ are the centroid coordinates, $t_j = j\Delta t \left(j = 1, \ldots, N\right)$, $\Delta t = 0.5$ s, and $N = 6000$ is the number of frames continuously recorded during the experiment. We defined the following quantities that were used in our analysis.

Velocity:

$$\vec{V}\left(t_j\right) = \frac{\vec{R}\left(t_j\right) - \vec{R}\left(t_{j-1}\right)}{\Delta t}$$

Heading:

$$\vec{H}\left(t_j\right) = \frac{\vec{V}\left(t_j\right)}{s\left(t_j\right)}$$

Scalar speed:

$$s\left(t_j\right) = \|\vec{V}(t_j)\|$$

Instantaneous turn rate:

$$\left|\frac{\Delta\theta}{\Delta t}\right|\left(t_j\right) = \frac{cos^{-1}\left(\vec{H}\left(t_{j-1}\right) \cdot \vec{H}^T\left(t_j\right)\right)}{\Delta t}, \left(0 \leq \Delta\theta \leq \pi\right)$$

Next, the Ramer–Douglas–Peucker algorithm (https://pypi.org/project/rdp/) was used to simplify the larval trajectories and therefore identify the locations where larvae executed turns. After visual inspection of the simplified trajectories, we fixed the distance dimension $\varepsilon$, that represents the maximum distance between the original points and the simplified curve. $\varepsilon = 2.5$ mm to the analysis with agar, sucrose, and apple juice and $\varepsilon = 1.25$ mm to the yeast analysis.

With turning points identified in the trajectory, the turning angles were obtained in the range $\left[-\pi, \pi\right]$ using the atan2 function in python. As a convention, clockwise turns were in the range $\left[-\pi, 0\right]$ and counter-clockwise turns in the range $\left[0, \pi\right]$. The handedness index of each larva was obtained as:

$$H = \frac{N_{CCW}}{N_{CCW} + N_{CW}}$$

where $N_{CCW}$ is the number of counter-clockwise turns in the trajectory and $N_{CW}$ the number of clockwise turns. Thus, if $H > 0.5$ ($H < 0.5$) the larva has a bias to execute more counter-clockwise (clockwise) turns.

From the turning points identified by the RDP algorithm, we built a vector that registers 1 in the time points where turns were registered and 0 otherwise. The length of this vector is the number of frames in the recording. Next, we applied a rolling window of 120 frames (1 min) to this vector and summed the elements within the window. Then, we averaged the number of turns registered within each 1-min window to obtain the average number of turns per minute.

## Patch radius and center coordinates

We used imageJ to determine the center and radius of each patch in the experiments. A frame of the recording was adjusted for contrast and brightness until the borders of the patch became visible. Circular regions of interest were drawn for each patch and the center coordinates and radius were obtained.

## Classification of turns as toward the patch center

Let $\vec{S}(t_k)$ be the trajectory simplified by the RDP algorithm, where each point is a turning point of the original trajectory. To classify the $k$ th turn in the trajectory as inwards or outwards, we define the following vectors:

$$\vec{V}_1 = \vec{S}(t_k) - \vec{S}(t_{k-1})$$
$$\vec{V}_2 = \vec{S}(t_{k+1}) - \vec{S}(t_k)$$
$$\vec{U} = \vec{P} - \vec{S}(t_k)$$

where $\vec{S}(t_{k-1})$ and $\vec{S}(t_{k+1})$ are the previous and the following turning locations and $\vec{P}$ is the center of the patch that is closest to $\vec{S}(t_k)$. The following angles are then computed (**Figure 3I**, left):

$$\theta_1 = \cos^{-1}\left(\frac{\vec{V}_1 \cdot \vec{U}}{\|\vec{V}_1\| \|\vec{U}\|}\right)$$
$$\theta_2 = \cos^{-1}\left(\frac{\vec{V}_2 \cdot \vec{U}}{\|\vec{V}_2\| \|\vec{U}\|}\right)$$

and the turn at $\vec{S}(t_k)$ is classified as inwards (outwards) if $\theta_2 < \theta_1$ ($\theta_2 > \theta_1$) (**Tao et al., 2020**).

## Model

### Homogeneous substrate

The simulated crawling substrate has rigid boundaries and the same dimensions as the behavioral arenas used in the experiments (240 × 240 mm²). At each time step $t_k$ the simulated larva can be at one of three different states (**Figure 2A**):

1. with probability $P_{crawl}$ crawling with speed $v(t_k) > 0$ sampled from a normal distribution;
2. with probability $P_{turn}$ turning an angle $\theta(t_k)$ sampled from a von Mises distribution;
3. with probability $P_{pause}$ paused ($v(t_k) = 0$).

The parameter values and distributions were obtained from our experimental data of larval crawling in homogeneous substrates and are unique for each type of larva and substrate (**Table 2**). Crawl, turn or pause events were registered with a constant probability per time step ($P_{crawl} = 1 - (P_{turn} + P_{pause})$) and the simulation duration was the same as our behavioral recordings (50 min). To capture the variability in the turning behavior, each larva was simulated with its own set of parameters for the turning angle distribution according to one recorded larva (with an average of 30 sitter and 30 rover larvae recorded at each type of substrate). The RDP algorithm was then used to identify salient turning points in the simulated trajectory (**Figure 2B**).

**Table 2.** Parameters of model in homogeneous and patchy substrates obtained in homogeneous substrate experiments.

| Substrate | Larva | Mean $v$ (mm/s) | Std $v$ (mm/s) | $P_{turn}$/s | $P_{pause}$/s |
|---|---|---|---|---|---|
| Agar | Rover | 0.84 | 0.13 | 0.044 | 0.0083 |
| Agar | Sitter | 0.96 | 0.13 | 0.046 | 0.0063 |
| Sucrose | Rover | 0.68 | 0.092 | 0.041 | 0.012 |

*Table 2. Continued on next page*

*Table 2. Continued*

| Substrate | Larva | Mean $v$ (mm/s) | Std $v$ (mm/s) | $P_{turn}$/s | $P_{pause}$/s |
|---|---|---|---|---|---|
| Sucrose | Sitter | 0.68 | 0.085 | 0.035 | 0.021 |
| Yeast | Rover | 0.37 | 0.13 | 0.033 | 0.25 |
| Yeast | Sitter | 0.31 | 0.11 | 0.028 | 0.25 |

## Patchy substrate

### Without biased turns toward the food

We modeled patchy environments initially with two circular patches (radius 25 mm) of food substrate (sucrose or yeast) with agar substrate in the rest of the arena (*Figure 2C*). Crawling speed and probabilities to turn or pause were drawn based on the current position of the simulated larva. The parameters were sampled from the corresponding food experiment when the larva was inside a patch, and sampled from the agar experiment when the larva was outside the patch. The turning angle distribution of each simulated larva corresponded to one from the recordings in the agar substrate. The same turning angle probability distribution was used whether the larva is inside or outside the patch. The initial position was picked at random in each simulation, but always inside one of the two food patches to match the experiments.

### With biased turns toward the food

Except for the choice of turning angles, the model was the same as the one described above. The biased choice of turns toward the food followed the implementation in *Tao et al., 2020*. After drawing a turning angle from the von Mises probability distribution, the turn direction was chosen such that the larva points toward the patch center with probability $P_{bias}$ that depends on the distance between the current position relative to the center of the closest patch (*Figure 5B*). When the simulated larva was further than 60 mm away from the closest patch center, no bias was applied in the turning direction since the data were very sparse in this region (most larvae never crawled such long distances away from the patch of food in the experiments). Each turn was defined by a set of three points $\{p_1, p_2, p_3\}$ where $p_1$ is where the turn initiates, $p_2$ is the end location of a left turn, and $p_3$ the end location of a right turn. Three movement vectors that characterize the turn options (to the left or to the right) were defined as:

$$\vec{v_1} = p_2 - p_1$$
$$\vec{v_2} = p_3 - p_1$$
$$\vec{u} = -p_1$$

We next calculated the angle $\theta$ the larval trajectory makes with the inward vector $\vec{u}$ when turning to the left ($p_2$) or to the right ($p_3$). The inward turn is the turn that results in the smallest $\theta$ (as shown in *Figure 5A*).

### With more patches

We fixed the total surface area of food to be distributed in $N$ patches as $S = 2\pi R^2$, where $R = 25$mm is the radius of the patches from the previous simulations and experiments. Then, the radius of each $N$ th patch is given by $R' = \sqrt{S/N\pi}$. The simulated larvae started within a random food patch, and were tracked for 50 min. The simulation parameters were kept the same as in the two patches model, except that the distances in the distance-dependent probability to turn inwards were adjusted for smaller patch radius, by multiplying the distance values by $R'/R$.

## Model parameters (Tables 2–4)

**Table 3.** Parameters of corrected model in patchy substrates obtained in patchy substrate experiments.

| Patchy substrate | Larva | Mean $v$ inside (outside) (mm/s) | Std $v$ inside (outside) (mm/s) | $P_{turn}$/s inside (outside) | $P_{pause}$/s inside (outside) |
|---|---|---|---|---|---|
| Agar + sucrose | Rover | 0.60 (0.65) | 0.27 (0.25) | 0.037 (0.044) | 0.039 (0.0014) |
| Agar + sucrose | Sitter | 0.52 (0.57) | 0.28 (0.26) | 0.030 (0.030) | 0.088 (0.040) |
| Agar + yeast | Rover | 0.37 (0.44) | 0.19 (0.20) | 0.039 (0.068) | 0.17 (0.023) |
| Agar + yeast | Sitter | 0.26 (0.36) | 0.14 (0.17) | 0.025 (0.048) | 0.32 (0.053) |
| Agar + apple juice | Rover | 0.44 (0.53) | 0.24 (0.26) | 0.026 (0.017) | 0.096 (0.076) |
| Agar + apple juice | Sitter | 0.39 (0.48) | 0.21 (0.22) | 0.021 (0.031) | 0.13 (0.065) |

**Table 4.** Statistical power and Cohen's effect size of non-significant comparisons.

**Figure 1**

| D – Avg. number of turns per min | | | Power ($1 - \beta$) | Cohen's size effect ($d$) |
|---|---|---|---|---|
| Rover | Agar | Sucrose | 0.24 | 0.34 |
| Rover | Sucrose | Yeast | 0.48 | 0.52 |
| Sitter | Sucrose | Yeast | 0.47 | 0.51 |
| *G – Handedness* | | | | |
| Rover | Agar | Sucrose | 0.13 | −0.22 |
| Rover | Agar | Yeast | 0.05 | 0.01 |
| Rover | Sucrose | Yeast | 0.19 | 0.31 |
| Sitter | Agar | Sucrose | 0.14 | 0.25 |
| Sitter | Agar | Yeast | 0.08 | 0.16 |
| Sitter | Sucrose | Yeast | 0.08 | −0.15 |

**Figure 3**

| C – Crawling speed | | | Power ($1 - \beta$) | Cohen's size effect ($d$) |
|---|---|---|---|---|
| Sucrose | | | | |
| Rover | In | Out | 0.14 | −0.34 |
| Sitter | In | Out | 0.20 | −0.44 |
| D – Avg. number of turns per min | | | | |
| Sucrose | | | | |
| Rover | In | Out | 0.29 | −0.56 |
| Sitter | In | Out | 0.05 | −0.02 |
| F – Handedness | | | | |
| Rover | | | | |
| Sucrose | In | Out | 0.11 | −0.30 |
| Yeast | In | Out | 0.10 | −0.24 |
| Apple juice | In | Out | 0.15 | 0.31 |
| Sitter | | | | |

*Table 4. Continued on next page*

*Table 4.* Continued

**Figure 1**

| Sucrose | In | Out | 0.06 | −0.15 |
|---|---|---|---|---|
| Yeast | In | Out | 0.06 | −0.14 |
| Apple juice | In | Out | 0.16 | −0.39 |
| *G – Crawl dist. 5 min* | | | | |
| Apple juice | Rover | Sitter | 0.20 | 0.34 |
| *H – Fraction of time spent inside patch* | | | | |
| Sucrose | Rover | Sitter | 0.08 | 0.20 |
| Apple juice | Rover | Sitter | 0.32 | −0.40 |

Figure S3

| *Apple juice Crawling speed* | | | Power $(1 - \beta)$ | Cohen's size effect ($d$) |
|---|---|---|---|---|
| In | Rover | Sitter | 0.61 | 0.60 |
| Out | Rover | Sitter | 0.32 | 0.40 |
| *Avg. number of turns per min* | | | | |
| In | Rover | Sitter | 0.53 | 0.55 |
| Out | Rover | Sitter | 0.09 | 0.18 |
| *Fraction of pauses* | | | | |
| In | Rover | Sitter | 0.32 | −0.40 |
| Out | Rover | Sitter | 0.07 | 0.10 |

**Figure 4**

| *Anosmic B – Crawling speed* | | | Power $(1 - \beta)$ | Cohen's size effect ($d$) |
|---|---|---|---|---|
| Sucrose | In | Out | 0.51 | −0.56 |
| Yeast | In | Out | 0.23 | −0.39 |
| *C – Avg. number of turns per min* | | | | |
| Sucrose | In | Out | 0.10 | −0.19 |
| Yeast | In | Out | 0.32 | −0.50 |
| *E – Handedness* | | | | |
| Sucrose | In | Out | 0.10 | 0.24 |
| Yeast | In | Out | 0.07 | −0.16 |
| Apple juice | In | Out | 0.45 | 0.58 |
| *F – Fraction of time spent inside patch* | | | | |
| | Sucrose | Yeast | 0.26 | 0.40 |
| | Sucrose | Apple juice | 0.44 | −0.50 |

**Figure 6**

| *C – Fraction of time spent inside patch – eight patches* | | | Power $(1 - \beta)$ | Cohen's size effect ($d$) |
|---|---|---|---|---|
| Sucrose | Rover | Sitter | 0.14 | −0.31 |
| Yeast | Rover | Sitter | 0.31 | 0.45 |
| *E – Fraction of visited patches* | | | | |

*Table 4.* Continued on next page

*Table 4. Continued*

**Figure 1**

| Rover | | | | |
|---|---|---|---|---|
| Sucrose | 2 patches | 8 patches | 0.05 | 0 |
| Sitter | | | | |
| Sucrose | 2 patches | 8 patches | 0.07 | 0.15 |
| Yeast | 2 patches | 8 patches | 0.56 | −0.62 |

## Acknowledgements

MW was supported by a Capes-Humboldt postdoctoral fellowship. JG was funded by the Max Planck Society. JB was funded by a Sir Henry Dale fellowship from the Wellcome Trust and Royal Society 105568/Z/14/Z. The authors thank Dr. Bertram Gerber and Michael Schleyer for helpful discussions during early stages of this work and also Andre Maia Chagas for technical assistance, Omar D Perez for statistical recommendations, Nick Humphries, Marla Sokolowski, Carlotta Martelli, and Alex Kacelnik for constructive comments on the manuscript.

## Additional information

### Funding

| Funder | Grant reference number | Author |
|---|---|---|
| Royal Society | 105568/Z/14/Z | Jimena Berni |
| Wellcome Trust | 105568/Z/14/Z | Jimena Berni |
| Max-Planck-Gesellschaft | | Marina E Wosniack Julijana Gjorgjieva |
| Alexander von Humboldt-Stiftung | | Marina E Wosniack |

The funders had no role in study design, data collection, and interpretation, or the decision to submit the work for publication. For the purpose of Open Access, the authors have applied a CC BY public copyright license to any Author Accepted Manuscript version arising from this submission.

### Author contributions

Marina E Wosniack, Conceptualization, Software, Formal analysis, Funding acquisition, Validation, Investigation, Visualization, Methodology, Writing - original draft; Dylan Festa, Formal analysis, Writing – review and editing; Nan Hu, Formal analysis, Investigation; Julijana Gjorgjieva, Conceptualization, Supervision, Funding acquisition, Validation, Methodology, Project administration, Writing – review and editing; Jimena Berni, Conceptualization, Formal analysis, Supervision, Funding acquisition, Validation, Investigation, Visualization, Methodology, Project administration, Writing – review and editing

### Author ORCIDs

Marina E Wosniack http://orcid.org/0000-0003-2175-9713
Dylan Festa http://orcid.org/0000-0003-3803-1542
Julijana Gjorgjieva http://orcid.org/0000-0001-7118-4079
Jimena Berni http://orcid.org/0000-0002-5068-1372

### Decision letter and Author response

Decision letter https://doi.org/10.7554/eLife.75826.sa1
Author response https://doi.org/10.7554/eLife.75826.sa2

## Additional files

### Supplementary files
- Transparent reporting form
- Source data 1. All data individual larvae for *Figures 1, 3, and 4*.
- Source data 2. All data individual larvae for *Figure 6*.

### Data availability
All data generated or analyzed during this study are included in the manuscript and supporting files 1 and 2. Source data files have been provided formal experimental data: Figures 1, 3, 4 and 6. Code available on GitHub: https://github.com/comp-neural-circuits/adaptation-of-drosophila-larva-foraging (copy archived at swh:1:rev:d501cd1f0df3df2fda0f286d58e98ee0f72b4898).

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
