## [Editor Report]

This paper contributes to the growing body of literature that investigates foraging in complex sensory landscapes. It is therefore of interest to both neuroscientists and ecologists. Using behavioral analysis and computational modeling, the authors characterize different behavioral components of the foraging strategy adopted by the *Drosophila* larva as a function of food quality and food distribution. Altogether, this works sets the stage for investigating the genetic and neural-circuit bases underlying the control of foraging behavior.

---

## [Decision Letter]

**Decision letter after peer review:**

Thank you for submitting your article "Adaptation of *Drosophila* larva foraging in response to changes in food resources" for consideration by *eLife*. Your article has been reviewed by 3 peer reviewers, and the evaluation has been overseen by a Reviewing Editor and K VijayRaghavan as the Senior Editor. The following individual involved in the review of your submission has agreed to reveal their identity: Mason Klein (Reviewer #4).

Essential revisions:

1. Defining a potential contribution of taste to foraging. Is it known that yeast is 'unnoticeable' by taste? Did the authors try or consider testing the Gr43a mutant on these substrates? When the larvae are pausing, are they trying to eat the substrate?

2. Defining the role of other modalities: On patchy substrates, is the border completely smooth or could the larvae also sense the border as a rough edge? Are there effects from mechanosensation due to the different preparation of the patch edge (printing for yeast versus cutting for juice and sucrose)? Does diffusion of patch edge play a role for apple juice and sucrose?

3. It would be important to highlight the contribution of the model. Could the model be used to guide new experiments or to suggest new ideas? Would the integration of chemotaxis in the model improve the reproduction of specific behavioral features observed experimentally? Related to this point, one aspect where the work could be strengthened is by highlighting or speculating about where the differences in patch residence times between the model and data might arise.

4. Technical concerns related to the methodology:

– Define whether the simulation's initial conditions matter.

– Specify whether the model samples a steady-state/ ergodic distribution of trajectories or if time-dependent properties should be accounted for.

– Compared to other models, no patch depletion is taken into account. Wouldn't this affect the leaving rates in small patches especially and more so than larger patches?

5. Statistics: For an animal that tends to have a very high variance in its behavior, the number of larvae used in each experiment seems to be pretty low. This limitation should be acknowledged in the conclusions drawn in the paper. More specifically, low sample sizes can lead to false negatives. To address this concern, the authors should perform a statistical power analysis to increase the confidence in the "not statistically significant" results.

The reviewers agreed that points 3 and 4 could be mainly addressed through computational work and/or editing. If any of the other concerns can be adequately addressed without additional experimental work, we will generally support this option so that it reduces the efforts entailed by the revision.

*Reviewer #2 (Recommendations for the authors):*

I have a few suggestions that I believe could strengthen the manuscript:

– Please comment on the simulation's initial conditions and if they matter? From the trajectories shown in Figures 2 and 5, it seems like there could be some finite duration effect

– Are you sampling a steady-state/ergodic distribution of trajectories or are there time-dependent properties?

– Could you show how well single-trial simulation data matches single-trial experimental data with respect to trajectory features?

It would be helpful for the reader if the model predictions from Figure 2 could be articulated more explicitly in the section discussing Figure 2. It is mentioned later, but it would be useful to summarize these results before the next section of experiments.

– What are the navigational metrics in apple juice on homogeneous patches?

– Does diffusion of patch edge play a role for apple juice and sucrose? Or are there effects from mechanosensation due to the different preparation of the patch edge (printing for yeats versus cutting for juice and sucrose)?

– Please explain the exact nature of the defect in anosmic larvae. The results appear somewhat confusing otherwise. Is it known that yeast is 'unnoticeable' by taste?

– Compared to other models, no patch depletion is taken into account. Wouldn't this affect the leaving rates in small patches especially and more so than larger patches? Relatedly, the paper could benefit if the phenomenological results of the model could be connected to optimal foraging theory, as the authors cite the marginal value theorem. However, this might be beyond the scope of this work.

*Reviewer #3 (Recommendations for the authors):*

I like the paper overall, the main results are interesting and well supported. I have some issues with the logic of the simulations and how they inform the experiments. My only "major" change would be to rearrange the paper a bit and put the simulations after what is now Figure 3, maybe even after all of the experimental data figures.

General/broader comments:

(1) Using 30 or fewer larvae for each experiment type feels quite low in my experience. I'm not suggesting performing a bunch of new experiments (I almost never do in reviews), but I think it's important to be careful with claims about any comparison being "the same" (specific instances pointed out below) -- finding "n.s." does not mean two things are the same, especially when a small number of data points is used in the comparison.

(2) The introduction feels like it's missing at least a whole paragraph that should probably be explaining to a non-fly, non-foraging researcher why this study is important and useful. I study fly larvae too, and it's certainly exciting and interesting to me, but I think some more effort to draw in other researchers is warranted.

(3) The simulations. I like the method of using results from actual experiments to make the probability distributions that the simulation draws from in its random-walk-style tracks. I'm sure the simulations are done properly and make sense. What dampens my enthusiasm for them in this specific paper is that they don't seem to be informing the experiments or suggesting new ideas to the extent that they could. Numerous statements that start with "as predicted by the model" make it sound like the simulations are providing some kind of insight, but often that doesn't seem true -- like the model predicting the animals spend more time in yeast patches vs. other food, of course they do, because the model makes them crawl slower and pause more in yeast. It's a cool result, but it comes from your real experiments. It's just not very clear what benefit the simulations have here. I don't mean they aren't useful or you should remove them, I mean it should be much more clear what they are for:

– they are used as a check to confirm what is very strongly implied by your own experiments. If you include the modulations from crawling in isotropic substrates, and you include chemotaxis, and then the simulations match your experiments pretty closely, then you have successfully identified the important behavioral features, right? –

I like studying larva behavior and I like building simulations of larva behavior too, but something about the flow and logic of how they are deployed here feels off. I'll try to be more specific in the detailed comments.

(4) When the larvae are pausing, are they trying to eat the substrate? It seems like with third instar larvae and a pretty low % gel, they could if they wanted to.

(5) You talk about distance traveled throughout the paper, but I don't think mention displacement. Wouldn't that be important for describing motion too? Like the animals diffusing away from their start point vs. traveling a lot of distance, these aren't quite the same thing when exploring an environment for food. In particular, when you mention reduced turn and increased pause rate -- don't these do opposite things? Turning less frequently means the animals leave a space faster and the higher pause rate keeps them there longer right? They are often mentioned together in the paper, with the net effect being that larvae spend longer times in such regions, and I trust that's true, but that's because the pausing rate (and duration of pauses) carries more weight in this case?

(6) Does what the larvae do in between turns (or pauses) matter? The simulations draw them as straight lines, but don't larvae drift towards preferable odors? (I could be remembering that wrong, but I think I saw that in a Louis lab paper at some point).

(7) Is there a reason the simulations are never referred to as Monte Carlo simulations, or as (modified) random walks? Those are two pretty important reference points, especially for a general audience I think?

More detailed comments:

(1) line 26: what does "permanence time" mean? The time they spend in the patches?

(2) line 35: "all living organisms need to explore their surroundings…" That doesn't sound true. Not all living organisms even move. And some animals in the ocean just sit around and let food come to them.

(3) line 53: "gradients of light"  "gradients of light intensity"? (light has a lot of properties).

(4) line 53: the paper used as a thermotaxis reference isn't really a thermotaxis paper, if you want one from the same group I would choose Luo, Gershow, et al. 2010, J. Neurosci. (For the record, I am an author on the one you already cite, not on the one I'm suggesting you add).

(5) line 94-95. This sentence is hard for me to understand.

(6) line 95-98. This is stated several times (basis for neuronal circuits), but is it explained why this is true? (in the Discussion section?)

(7) line 103-104. This section title maybe needs a comma and "pauses frequency" should say something else?

(8) line 113. 50 minutes seems like a long time for these experiments. If I'm back-of-the-enveloping something moving a 1 mm/s, turning every 20 s… wouldn't they leave the arena a lot faster than that? Does the arena have walls? Do their pauses last a really long time on average?

(9) Figure 1.

In 1B. Could the pauses be marked too? Maybe with open/empty circles instead of a solid/filled one? The pausing seems so important in this paper, it seems weird to leave it out.

In 1A, could there be labels for the camera and the red thing (the IR filter?).

In 1B (this figure and others), I think you mean 3pi/2 for the lower quadrant in the turn angle distributions, not 2pi/3?

Also, I don't think you need the same scale bar three times for the three panels.

Could it maybe be more clear that 1F is kind of a summary/consequence of C/D/E? It's kind of the main summary statistic here.

Some of the panels have four * symbols, but the p-value for that is not given in the caption (this figure and others).

1D: (here and elsewhere) when the turning rate is calculated, I get that turns in just the number of turns, but what's in the denominator? The actual time elapsed, or only the time during forward crawling (i.e., the total time when they are could start a turn, excluding when they are already turning or paused).

Maybe the panels and overall figure could be a bit larger? It's a little hard to read at this size.

(10) line 156. would it make sense to do a significance test for your strong handedness? The very long experiments help you here because you measure a lot of turns for individuals, but presumably, a 50/50 binomial distribution of turns would yield some "strong handedness" animals too right?

(11) line 164-167. Are you sure about that? Your data in S1F might turn out to be significant with more data taken? Similarly for the yeast substrate in Figure S1B (mentioned in line 223). I would hesitate with claims about non-significance when comparing a pretty small number of larvae.

(12) Figure 2

Why would only 30 larvae be simulated? It's a simulation, you can simulate millions of them if you wanted?

(13) Line 226ff. This is where the simulation logic seems weird to me. Maybe the order of the sections in the paper throws it off. This section laid out a simulation method, then ran simulations with patches of food substrates. But at this point you know the simulation is missing chemotaxis, and you know that real animals can smell things and move towards/away from them. I'm not sure what the point is of setting this up and then "testing the predictions of the model" with your (very cool) food patch experiments. Similarly in line 281ff, saying "the model predicted" that larvae spend more time in certain substates, doesn't feel like the right statement, because the model is only telling you what you directly put in yourself. This part is written as if the model is helping you figure out what's going on, but that doesn't seem true in this part. I think it might be better to put the simulation stuff *after* you have looked at the experimental patch data, then build up its capabilities if you want, or just include chemotaxis right away. The in between thing with only using the isotropic data kind of feels like a waste of time in this paper, and really interrupts the flow of some really interesting experimental results.

(14) line 232-233. Also, an oddly phrased section heading.

(15) line 237. What is the reason for apple juice showing up here suddenly? The other substances seem more fundamental -- what chemicals are in apple juice that makes it attractive? Why weren't there isotropic experiments done with an apple juice substrate?

(16) Figure 3.

3B, could this have a legend that says what the white and black circles are? (I know it's in the caption, but it would help to put it in both places).

3I. Could this be drawn more clearly with labels? (the part defining the angles). To understand what you are doing here I had to read the methods section, draw my own picture, then go look up the Tao reference (which has a better picture). Also, a general issue/concern with this definition of "towards the center". Wouldn't there be many circumstances where *either* a right or a left turn (of the same size) would point the animal more towards the center? Is the *rate* that the larvae turn different as they crawl away from a food source? Would anything change if you defined "towards the center" as the larva picking the direction that points them *more* toward the center than the other direction? It seems like some crawling directions would have more crucial turning decisions in them, like crawling perpendicular to the vector from food patch center (matters the most) vs. crawling parallel (directly away from the food, doesn't matter which way you turn). Pooling the turn decisions into those subsets (like you do with distance away) might make the effect more pronounced.

(17) line 300. I think you mean J instead of B?

(18) line 304. Why is this surprising? Larvae can smell, you cited a bunch of papers earlier that show this in detail.

(19) line 329 + Figure 4 -- again, a claim about non-significance with a small data set is maybe not be warranted.

(20) line 384. "Therefore…" Again, this implies the simulation result is telling you something, but you already knew this before simulating, based on the real anosmic data.

(21) line 390-392, this seems like a really interesting idea -- could this be expanded/discussed further in the discussion part of the paper?

(22) line 426ff. I think it would help to more clearly state which simulation results are obvious based on what you put in the simulation, and which gives you something unexpected.

(23) line 480-482. Here too, larvae spending less time in less nutritious patches, isn't that a direct result of putting your empirical result into the simulation? They crawl faster and pause less, doesn't this have to be true?

(24) line 636: is that hours after egg laying or after eclosion?

(25) line 641ff: I don't quite follow -- you are choosing a lower frame rate in order to prioritize spatial resolution because the camera doesn't run faster when recording at 2048x2048?

(26) line 690: when is instantaneous turn rate used? You are generally finding the total for long trajectories right?

(27) line 695: What is the distance dimension parameter? Could you briefly explain what it means?

(28) line 699: typo with the brackets vs. parentheses.

---

## [Author Response]

Essential revisions:1. Defining a potential contribution of taste to foraging. Is it known that yeast is 'unnoticeable' by taste? Did the authors try or consider testing the Gr43a mutant on these substrates? When the larvae are pausing, are they trying to eat the substrate?

Foods are complex in nature: they have different texture, odour and taste. The goal of our paper was not to investigate how different sensory modalities affect the foraging strategy, but rather to understand how different foods, with their combined properties, modulate foraging. Taste is however a key sensory input that contributes to the decision of an animal to feed, so it is indeed interesting to discuss its role in the behavioural adaptation we observed. We have added a new section where we discuss how taste and smell contribute to the perception of food including yeast.

Line 634: “An experiment using the gustatory sweet sensor *Gr43a* mutant on sucrose, which is not volatile and does not produce smell, could help discerning the contribution of taste at the border of the patch (Fujishiro et al. 1984; Marella et al., 2006; Miyamoto et al. 2013; Wang et al.,2004; Mishra et al.,2013). For yeast, the lack of smell completely changed the response of the larvae, which did not show differences inside and outside the patch for most foraging parameters (Figure 4B, C, E, G). In this instance, taste was not sufficient to retain larvae inside the yeast patch (compare Figure 3H with Figure 4F) even though several gustatory receptors have been shown to be activated by yeast metabolites (Wisotsky et al., 2011, Ganguly et al.,2017, Croset et al., 2016).”

To address the question of what the larvae do during the pauses, we recorded high magnification movies of larvae that paused extensively in yeast. We found that the majority of these larvae were actually completely motionless and only showed movements of food in their gut. We have added a description and video 1 in results, line 165: “Most pausing larvae were completely still, except for internal movements in their gut, suggesting they were digesting (Video 1).”

In discussion, line 579: “They also made more pauses, with the majority remaining stationary, except for internal gut movements (Video 1), which suggested that they were digesting the yeast. The pauses were rarely observed in sucrose, which is metabolized more quickly than yeast, even when mixed with agar (Figure 1E).”

2. Defining the role of other modalities: On patchy substrates, is the border completely smooth or could the larvae also sense the border as a rough edge? Are there effects from mechanosensation due to the different preparation of the patch edge (printing for yeast versus cutting for juice and sucrose)? Does diffusion of patch edge play a role for apple juice and sucrose?

The point raised by the reviewers regarding the edge of the food is quite relevant. To test if the edge is sufficient to generate changes in behaviour comparable to what we observed with food, we performed new experiments and included two controls. First, we built the arena exactly as we did in the two patch experiments, but we filled the circular patches with agar only. In this situation there is a smooth edge between the “out” and “in” area. Second, we imitated the texture of the yeast by spreading a drop of ultrasound gel (see Methods, page 35 line 796). Although the viscosity is not exactly the same, this control experiment helps us understand how the larvae behave when crawling across an edge between a smooth and viscus substrate (the gel) and a flat and harder one (the agar).

The results have been added as Figure 3-supplement 2 and they show that: Line 339:

“To control for possible mechanosensory effects due to the border edges, we prepared new arenas with patches that contained no nutrients, either using the same agar that composed the rest of the arena, or using ultrasound gel (Methods). Larvae in the agaragar or the agar-gel border did not show any changes in their preference to turn inwards, towards the patch center, confirming that the behavioral change observed in response to food is specific (Figure 3—figure supplement 2).”

Diffusion is certainly happening at patch edges for sucrose and apple juice. To account for changes in behavior that depend on diffusion, we have also added an analysis of turn probability in anosmic larvae:

Line 408: “Taste very likely influences the probability that larvae remain in the patches. To control for the effect of diffusion of nutrients (sucrose and apple juice) at the edge of a patch, we evaluated the maximum distance at which an increased fraction of turns toward the center was significantly different when compared to the yeast non-responsive anosmic control. At a distance greater than 0.5 cm from the edge, anosmic larvae on sucrose, apple juice, and yeast were indistinguishable, suggesting that diffusion has a limited impact on behavior (Figure 4, figure supplement 1)”

And a paragraph in discussion, line 629: “In anosmic larvae, the fraction of turns towards the center for sucrose and apple juice patches was only higher compared to the one for the yeast patch (where there was no food effect) within the first half centimeter outside the patch, suggesting that the impact of diffusion could be significant only in that region (Figure 4—figure supplement 1). ”

In addition to this, to control for the effects of diffusion over time, we added a new analysis and a new supplementary figure.

Line 415: “Finally, to control for possible effects of diffusion over time, we compared the fraction of turns towards the center in the first and second half of the experiment. For most distance and nutrients, the two distributions were not significantly different (Figure 4—figure supplement 1B).

3. It would be important to highlight the contribution of the model. Could the model be used to guide new experiments or to suggest new ideas? Would the integration of chemotaxis in the model improve the reproduction of specific behavioral features observed experimentally? Related to this point, one aspect where the work could be strengthened is by highlighting or speculating about where the differences in patch residence times between the model and data might arise.

The model plays a crucial role in this study. Its first advantage is to extract essential features that explain how larvae move across different environments. Second, the model enabled us to simulate a much wider range of conditions, parametrized by food fragmentation level, than possible experimentally (Figure 5E-G). For example, although we originally developed the model based on the 2-patch experiments, the model predicted larval behavior on 8-path experiments without further adjustments which was then confirmed in real larvae. Therefore, the model was already used to guide new experiments.

See for e.g. line 538: “As predicted by the model, both rovers and sitter spent half of the time inside patches when the area of food was divided in eight compared to two patches (Figure 5F and 6C,D).”

Our model is phenomenological rather than mechanistic. Although it does not explicitly include chemotaxis, by incorporating “biased turns” where the probability of turning towards a patch depends on the distance from its center, the model already implicitly includes the effect of chemotaxis. To capture further aspects of the data (including patch residence times) will likely require a more detailed mechanistic model where different sensory modalities are included, as opposed to our phenomenological model based on behavioural statistics. This is very interesting but beyond our scope (and goals). Nonetheless, our phenomenological model can be used as a term of comparison and benchmark for future bottom-up mechanistic models that could explicitly represent sensory information, internal variables (e.g. short-term memory traces, satiety level) and neural circuit components.

We have rephrased several sections to highlight the predictions of the model: Line 257: “We therefore compared the model predictions on foraging efficiency in patchy environments with behavioral experiments.”

Line 473: “We next used our model to investigate how a further fragmentation of the food patches affects the ability of larvae to stay in patches where they can feed.” […] “In total, this would represent 1575 hours of experiment, highlighting the advantage of the model.”

Line 651: “One of the strengths of our phenomenological model is that it incorporates a modular organization of foraging that could reflect how the crawl and turn modules are controlled. First, we modelled a stochastic search where no information regarding food is available outside of the current location, because food is absent or because the larvae cannot sense it. This corresponds to an autonomous search behavior implemented by circuits located in the ventral nerve cord without input from the brain (Berni et. al 2012; Sims et al. 2019). Second, we incorporated a goal-directed navigation that allows larvae to return to the food. Our phenomenological model includes a distance-dependent probability to turn inwards that mimics the effect of chemotaxis (when present), as much as any other possible mechanism that contributes to the turning probability. As a consequence, we observed that simulated larvae, even when the resources are fractioned in eight patches, could stay inside the food patch for longer periods, in line with experimental observations (Figure 5 and Figure 6). The model could be improved by setting the turning properties outside the patch to match as closely as possible experimental observations. To this end, we could consider studies of larvae crawling in different attractive gradients, where the changes in turning probability and angle, including weathervaning, have been investigated in relation to precise spatio-temporal information of odorants (Louis et al., 2008; Gomez-Marin et al., 2011; Davies et al.,2015). It would also be helpful to have information about other attractive gradients, like taste, to know if a common set of mechanisms is used regardless of the sensory modality. Using this information, our model could be used to investigate how crawling speed and turning properties are controlled via descending pathways from the brain (Tastekin et al. 2018; Jovanic et al. 2019). Finally, in the presence of nutrients, our model adjusts movements to stay on the food patch. The concerted decrease in turning rate and crawling speed and the increase in the number of pauses suggests that a neuromodulatory depression of movement (Marder, 2012) could be relevant in this phase. It would be interesting to investigate more generally how neuromodulators influence the decision to remain or explore new food resources in relation to the resources available and the larval motivational state.”

4. Technical concerns related to the methodology:– Define whether the simulation's initial conditions matter.

The initial conditions and the time duration of the simulation were chosen in line with experiments, because our main goal is to make a quantitative comparison between the two. A different choice of initial conditions, for example, starting outside of a food patch, could lead to different statistics since the area without food is larger than the patches with food, but this would require new experiments to make the comparison to the model. We added these details in the text:

Line 900: “The initial position was picked at random in each simulation, but always inside one of the two food patches to match the experiments.”

Line 925: “The simulated larvae started within a random food patch, and were tracked for 50 minutes.”

– Specify whether the model samples a steady-state/ ergodic distribution of trajectories or if time-dependent properties should be accounted for.– Compared to other models, no patch depletion is taken into account. Wouldn't this affect the leaving rates in small patches especially and more so than larger patches?

Even in the eight-patches experiments, the amount of food in a single patch cannot be depleted in the 50 minutes of recording. For this reason, effects of depletion or food degradation have not been included in the model.

5. Statistics: For an animal that tends to have a very high variance in its behavior, the number of larvae used in each experiment seems to be pretty low. This limitation should be acknowledged in the conclusions drawn in the paper. More specifically, low sample sizes can lead to false negatives. To address this concern, the authors should perform a statistical power analysis to increase the confidence in the "not statistically significant" results.

The low number of larvae is compensated by the 50 minutes recording time. We have added table 4 where we have calculated the power and Cohen’s size effect of all non-significant comparisons in our behavioural experiments.

The reviewers agreed that points 3 and 4 could be mainly addressed through computational work and/or editing. If any of the other concerns can be adequately addressed without additional experimental work, we will generally support this option so that it reduces the efforts entailed by the revision.Reviewer #2 (Recommendations for the authors):I have a few suggestions that I believe could strengthen the manuscript:– Please comment on the simulation's initial conditions and if they matter? From the trajectories shown in Figures 2 and 5, it seems like there could be some finite duration effect– Are you sampling a steady-state/ergodic distribution of trajectories or are there time-dependent properties?– Could you show how well single-trial simulation data matches single-trial experimental data with respect to trajectory features?

In simulations and in experiments, larvae started from a food patch, since larval eggs are laid on fruit, and then were tracked for the following 50 minutes. Rather than measuring the ergodic properties of the simulated walk, in our simulations (by investigating the effects of initial conditions and duration), we aimed to quantitatively compare the model to experimental data.

We now added this detail in the text:

Line 193: “In ecological conditions, the fruit on which *Drosophila* eggs are laid and on which the larvae forage decays over time.”

Line 900: “The initial position was picked at random in each simulation, but always inside one of the two food patches to match the experiments.”

Line 925: “The simulated larvae started within a random food patch, and were tracked for 50 minutes.”

Regarding single trials, Figure 6A and 6B show that simulated and measured paths resemble each other at a qualitative level. Due to the variability between animals, quantitative comparisons at the single-trial level require models based on single larvae, and not on average measures across many larvae as in our current model. The current model takes parameters such as pause duration and step length from the homogeneous experiments, whereas the bias in turning towards patch centre comes from the two patches experiment. Fitting a model on a single larva would therefore require a new experiment where the same individual is tracked in different arenas. We agree this is very interesting but leave it for future work.

It would be helpful for the reader if the model predictions from Figure 2 could be articulated more explicitly in the section discussing Figure 2. It is mentioned later, but it would be useful to summarize these results before the next section of experiments.

They have been clarified, thanks for the suggestion.

Line 252: “Thus far, our model predicts that, in patchy environments, larvae spend a relatively small proportion of time inside patches (approximately 1% for sucrose and 3% for yeast) while exploring takes up most of their time with a significant energy cost.”

– What are the navigational metrics in apple juice on homogeneous patches?

We did not perform experiments with apple juice in a homogeneous arena. We included apple juice only when we performed experiments with two and eight food patches. We now explain:

Line 268: “We also performed experiments using apple juice as a nutrient, motivated by the fact that it is ecologically relevant and that, unlike sucrose, the fructose contained in apple juice is volatile, which makes it detectable by smell and not only by taste.”

– Does diffusion of patch edge play a role for apple juice and sucrose? Or are there effects from mechanosensation due to the different preparation of the patch edge (printing for yeats versus cutting for juice and sucrose)?

To investigate the role of diffusion, we re-analysed the data by separating it into the first and second half of the experiment (Figure 4—figure supplement 1). To investigate the role of mechanosensation, we performed new control experiments to test the impact of the edge (Figure 3—figure supplement 1E and F) and calculated the fraction of turns towards the center in anosmic larvae. See above for details.

– Please explain the exact nature of the defect in anosmic larvae. The results appear somewhat confusing otherwise. Is it known that yeast is 'unnoticeable' by taste?

We have clarified what the defect in anosmic larvae is, by explaining the Orco receptor. Line 368:

“Thus, we repeated the patchy experiments with mutant anosmic larvae, where Orco, the obligatory co-receptor for all olfactory neurons, apart the CO2 sensing ones, is mutated (Vosshall and Stocker, 2007) and tested if they show the same distant-dependent bias when exploring the patchy substrate.”

Metabolites from yeast are known to be sensed by taste but not enough information is available about the exact nature of sensing. We have added information in the discussion (see above).

– Compared to other models, no patch depletion is taken into account. Wouldn't this affect the leaving rates in small patches especially and more so than larger patches? Relatedly, the paper could benefit if the phenomenological results of the model could be connected to optimal foraging theory, as the authors cite the marginal value theorem. However, this might be beyond the scope of this work.

Larvae eat relatively little compared to the amount of food available to them in the patches, even in the 8 patches arena, so we have not considered patch depletion in the model.

Adding a measure of optimality would require new analyses and model extensions, therefore we consider applications to optimal foraging theory beyond the scope of this

Reviewer #3 (Recommendations for the authors):I like the paper overall, the main results are interesting and well supported. I have some issues with the logic of the simulations and how they inform the experiments. My only "major" change would be to rearrange the paper a bit and put the simulations after what is now Figure 3, maybe even after all of the experimental data figures.General/broader comments:(1) Using 30 or fewer larvae for each experiment type feels quite low in my experience. I'm not suggesting performing a bunch of new experiments (I almost never do in reviews), but I think it's important to be careful with claims about any comparison being "the same" (specific instances pointed out below) -- finding "n.s." does not mean two things are the same, especially when a small number of data points is used in the comparison.

As noted above, the low number of larvae is compensated by the 50 minutes recording time. We note that this is much longer than what most other experiments tracking crawling larvae have achieved so far. We have added Table 4 where we calculated the power and Cohen’s size effect of all non-significant comparisons in our behavioural experiments. As expected, the power tends to be low, but it gives a better idea of the confidence of each comparison.

(2) The introduction feels like it's missing at least a whole paragraph that should probably be explaining to a non-fly, non-foraging researcher why this study is important and useful. I study fly larvae too, and it's certainly exciting and interesting to me, but I think some more effort to draw in other researchers is warranted.

We have edited the text to make it more clear why our results apply beyond larval crawling (see response to reviewer 2 above)

(3) The simulations. I like the method of using results from actual experiments to make the probability distributions that the simulation draws from in its random-walk-style tracks. I'm sure the simulations are done properly and make sense. What dampens my enthusiasm for them in this specific paper is that they don't seem to be informing the experiments or suggesting new ideas to the extent that they could. Numerous statements that start with "as predicted by the model" make it sound like the simulations are providing some kind of insight, but often that doesn't seem true -- like the model predicting the animals spend more time in yeast patches vs. other food, of course they do, because the model makes them crawl slower and pause more in yeast. It's a cool result, but it comes from your real experiments. It's just not very clear what benefit the simulations have here. I don't mean they aren't useful or you should remove them, I mean it should be much more clear what they are for:– they are used as a check to confirm what is very strongly implied by your own experiments. If you include the modulations from crawling in isotropic substrates, and you include chemotaxis, and then the simulations match your experiments pretty closely, then you have successfully identified the important behavioral features, right? –I like studying larva behavior and I like building simulations of larva behavior too, but something about the flow and logic of how they are deployed here feels off. I'll try to be more specific in the detailed comments.

Since our model is purely phenomenological and directly based on the experimental measures, it cannot offer explicit insights into the mechanisms that determine the animal’s behaviour. However, the model still produces relevant, quantitative results that have been used to guide our experiments, and that can, in turn, inspire future work. As we answer in the required revisions, we used the model’s predictions from the two-patch experiments to predict the behavior for the eight-patch experiments without further adjustments. The model also highlights the consistency of behavioural parameters across conditions: the values measured in isotropic substrates could still be used in the patch experiments where chemotaxis effects exist.

We also added a new section in the discussion to emphasize the benefits of using the model (line 651, already referenced in the answer to the other reviewers).

(4) When the larvae are pausing, are they trying to eat the substrate? It seems like with third instar larvae and a pretty low % gel, they could if they wanted to.

Pausing larvae are generally still, except for gut movements. We included a supplementary video as an example of this behavior.

Line 165: “Most pausing larvae were completely still, except for movements of food in their gut suggesting they were digesting (Video 1).”

(5) You talk about distance traveled throughout the paper, but I don't think mention displacement. Wouldn't that be important for describing motion too? Like the animals diffusing away from their start point vs. traveling a lot of distance, these aren't quite the same thing when exploring an environment for food.

In this paper we did not consider displacement because it is not very informative in the patches experiments where the patch size restricts the distance larvae explore. We also noticed that when they leave the patch, larvae generally stay in proximity (maximum 6 cm from centre of patch) before returning (Figure 3).

The current model samples step sizes from a normal distribution based on the data. Accounting for large displacements would require different statistics, for example a heavytailed distribution, as in Lévy random walks. It would be interesting to extend the model in this direction, however, since larvae moved slowly and with frequent changes of directions, we did not consider it necessary in the current work.

In particular, when you mention reduced turn and increased pause rate -- don't these do opposite things? Turning less frequently means the animals leave a space faster and the higher pause rate keeps them there longer right? They are often mentioned together in the paper, with the net effect being that larvae spend longer times in such regions, and I trust that's true, but that's because the pausing rate (and duration of pauses) carries more weight in this case?

What unifies all these differences in behaviour is the fact that larvae move less. So indeed they reduce turn frequency but that doesn’t mean that they go farther. They reduce the turn frequency and crawling speed at the same time. As a result, they do not leave the patch of food on which they were placed. We have added a comment in discussion:

Line 675: “The concerted decrease in turning rate and crawling speed and increase in the number of pauses, suggests that a neuromodulatory depression of movement (Marder, 2012) could be relevant in this phase.”

(6) Does what the larvae do in between turns (or pauses) matter? The simulations draw them as straight lines, but don't larvae drift towards preferable odors? (I could be remembering that wrong, but I think I saw that in a Louis lab paper at some point).

Yes, the reviewer is correct: weathervaning has been described in larvae during chemotaxis (Gomez-Marin and Louis, 2014). It is an orientation mechanism that occurs as larvae crawl in a gradient. In our case, it could have influenced the behaviour at the border of the patch but we have not included it in the model since we have no information about the gradient the larvae are exposed to. We have mentioned it in discussion line 665 and see above.

(7) Is there a reason the simulations are never referred to as Monte Carlo simulations, or as (modified) random walks? Those are two pretty important reference points, especially for a general audience I think?

We preferred not to use the term “Monte Carlo”, because it might generate confusion on how we conducted our simulations. In particular, the results of our simulations are restricted to initial conditions set on food patches, and for a time of 50 minutes, in line with the experiments. Typical Monte Carlo approaches instead operate in an ergodic / steady-state regime (i.e. completely random initial conditions and time that tends to infinity).

Regarding random walks, they are now mentioned in the introduction.

Line 46: “In environments where resources are abundant, animals will search and exploit them performing short movements in random directions, in patterns well approximated by Brownian random walks. When resources are sparse, and foragers have incomplete knowledge about their location, a more diffusive strategy is needed, with an alternation between short-range and long-range movements, which can be modelled as a Lévy random walk. Analysis of animal movements in the wild has demonstrated that environmental context can induce the switch between Lévy to Brownian movement patterns (Humphries et al., 2010), but the effective mechanisms behind the implementation of such behavior (e.g., cognitive capacity, memory) often remain elusive (Budaev et al., 2019).”

However, we did not refer directly to our model as a “random walk”, since that would then require us to formalize and explain the aspects in which our approach differs from classical random walk processes (e.g. dependence on location, biases in heading).

More detailed comments:(1) line 26: what does "permanence time" mean? The time they spend in the patches?

This has been corrected.

(2) line 35: "all living organisms need to explore their surroundings…" That doesn't sound true. Not all living organisms even move. And some animals in the ocean just sit around and let food come to them.

Replaced by” most moving organisms”

(3) line 53: "gradients of light"  "gradients of light intensity"? (light has a lot of properties).

True, this has been corrected.

(4) line 53: the paper used as a thermotaxis reference isn't really a thermotaxis paper, if you want one from the same group I would choose Luo, Gershow, et al. 2010, J. Neurosci. (For the record, I am an author on the one you already cite, not on the one I'm suggesting you add).

Thank you, reference added.

(5) line 94-95. This sentence is hard for me to understand.

Rephrased by replacing “weight” by “degree” of the behavioural response.

(6) line 95-98. This is stated several times (basis for neuronal circuits), but is it explained why this is true? (in the Discussion section?)

We have added a section to explain this:

Line 671: “Using this information, our model could be used to investigate how crawling speed and turning properties are controlled via descending pathways from the brain (Tastekin et al. 2018; Jovanic et al. 2019).” But see the whole new paragraph.

(7) line 103-104. This section title maybe needs a comma and "pauses frequency" should say something else?

We have slightly changed the title so it now reads:

Line 119: “Food quality controls the distance travelled modulating the speed and the frequency of pauses”. We believe it is now clearer.

(8) line 113. 50 minutes seems like a long time for these experiments. If I'm back-of-the-enveloping something moving a 1 mm/s, turning every 20 s… wouldn't they leave the arena a lot faster than that? Does the arena have walls? Do their pauses last a really long time on average?

When feeding, larvae are not interested in leaving the arena. When searching in homogenous agar, they do not like the dry edges of the arena and stay inside.

(9) Figure 1.In 1B. Could the pauses be marked too? Maybe with open/empty circles instead of a solid/filled one? The pausing seems so important in this paper, it seems weird to leave it out.

We tried this, but the plot of trajectories becomes really crowded and it becomes very hard to make sense of the larval movement. On a more practical note, the first author of the paper is on maternity leave (why we added a second author to perform the analysis of the new experiments) and it is very difficult for her to explore other aspects of figure design now. We are sorry but we won’t be able to add them at this stage.

In 1A, could there be labels for the camera and the red thing (the IR filter?).

Thank you, done.

In 1B (this figure and others), I think you mean 3pi/2 for the lower quadrant in the turn angle distributions, not 2pi/3?

Yes, thank you, this has been corrected.

Also, I don't think you need the same scale bar three times for the three panels.

We wanted to emphasize that the scale was the same because the tracks look very different in length. We have only kept 3 scales in that figure.

Could it maybe be more clear that 1F is kind of a summary/consequence of C/D/E? It's kind of the main summary statistic here.

In line 166 we now write: “as a consequence”.

Some of the panels have four * symbols, but the p-value for that is not given in the caption (this figure and others).

This has now been corrected in all Figures

1D: (here and elsewhere) when the turning rate is calculated, I get that turns in just the number of turns, but what's in the denominator? The actual time elapsed, or only the time during forward crawling (i.e., the total time when they are could start a turn, excluding when they are already turning or paused).

We provide this information in the figure legend: “Average number of turns per minute registered in each trajectory”.

Maybe the panels and overall figure could be a bit larger? It's a little hard to read at this size.

We agree that it is very important for the reader to clearly see all aspects of the figures. We considered splitting each figure into multiple figures but decided against that because each figure relays its own message. We hope that since the journal publishes only online, the readers will be able to zoom into the different regions of each figure for details that are not easily seen in the printed version. We are making sure that the final figures have the maximum possible width to maximize readability.

(10) line 156. would it make sense to do a significance test for your strong handedness? The very long experiments help you here because you measure a lot of turns for individuals, but presumably, a 50/50 binomial distribution of turns would yield some "strong handedness" animals too right?

Yes, but we do not feel it is necessary here. A statistical test just confirms what we already see, and it is not important for the main message of the paper, which is focused on exploration strategies in different substrates.

(11) line 164-167. Are you sure about that? Your data in S1F might turn out to be significant with more data taken? Similarly for the yeast substrate in Figure S1B (mentioned in line 223). I would hesitate with claims about non-significance when comparing a pretty small number of larvae.

We have changed it accordingly:

Line 184: “Thus, when the resources are distributed homogenously, the genetic foraging dimorphism could not be detected.”

(12) Figure 2Why would only 30 larvae be simulated? It's a simulation, you can simulate millions of them if you wanted?

The idea was to generate a representative sample comparable to the actual animals using the information provided by the animals. As explained in the paper: “each simulated larva had its own set of parameters for the turning angle distribution based on a single recorded larva.”

(13) Line 226ff. This is where the simulation logic seems weird to me. Maybe the order of the sections in the paper throws it off. This section laid out a simulation method, then ran simulations with patches of food substrates. But at this point you know the simulation is missing chemotaxis, and you know that real animals can smell things and move towards/away from them. I'm not sure what the point is of setting this up and then "testing the predictions of the model" with your (very cool) food patch experiments. Similarly in line 281ff, saying "the model predicted" that larvae spend more time in certain substates, doesn't feel like the right statement, because the model is only telling you what you directly put in yourself. This part is written as if the model is helping you figure out what's going on, but that doesn't seem true in this part. I think it might be better to put the simulation stuff after you have looked at the experimental patch data, then build up its capabilities if you want, or just include chemotaxis right away. The in between thing with only using the isotropic data kind of feels like a waste of time in this paper, and really interrupts the flow of some really interesting experimental results.

We have rewritten the section. The experiments were designed and performed in line with the classical approach used in modelling of Levy flights (here foraging) where only the diffusion is considered when evaluating the foraging in different environments. It is clarified in line 198: “The model predicted the fraction of time larvae spent inside patches of food, as a measure of food exploitation, if larvae only used the information about the substrate while foraging“. The diffusion models can be really good to quantify encounters at a large scale (as previous work on Levy flights has shown) but clearly they are insufficient to understand exploitation.

(14) line 232-233. Also, an oddly phrased section heading.

This has been re-organised.

(15) line 237. What is the reason for apple juice showing up here suddenly? The other substances seem more fundamental -- what chemicals are in apple juice that makes it attractive? Why weren't there isotropic experiments done with an apple juice substrate?

We now explain that we added apple juice because we wanted a substance that was ecologically relevant and attractive to the larvae. This is the case because it contains fructose and smells.

Line 268: “We also performed experiments using apple juice as nutrient, motivated by the fact that it is ecologically relevant and that, unlike sucrose, the fructose contained in apple juice is volatile, which makes it detectable by smell and not only by taste.”

(16) Figure 3.3B, could this have a legend that says what the white and black circles are? (I know it's in the caption, but it would help to put it in both places).

Done

3I. Could this be drawn more clearly with labels? (the part defining the angles). To understand what you are doing here I had to read the methods section, draw my own picture, then go look up the Tao reference (which has a better picture). Also, a general issue/concern with this definition of "towards the center". Wouldn't there be many circumstances where either a right or a left turn (of the same size) would point the animal more towards the center? Is the rate that the larvae turn different as they crawl away from a food source? Would anything change if you defined "towards the center" as the larva picking the direction that points them more toward the center than the other direction? It seems like some crawling directions would have more crucial turning decisions in them, like crawling perpendicular to the vector from food patch center (matters the most) vs. crawling parallel (directly away from the food, doesn't matter which way you turn). Pooling the turn decisions into those subsets (like you do with distance away) might make the effect more pronounced.

Crawling is definitely more complex than just turning in one direction, but we were trying to find the simplest characterization to use in the model. The rate and angle of turns are likely to be highly dependent on gradients. We have a new section in discussion (see above). And we have re-drawn the figure.

(17) line 300. I think you mean J instead of B?

It is B but the sentence should say “shown in black.”

(18) line 304. Why is this surprising? Larvae can smell, you cited a bunch of papers earlier that show this in detail.

You are right, we deleted the word “Surprisingly”.

(19) line 329 + Figure 4 -- again, a claim about non-significance with a small data set is maybe not be warranted.

We have added the power analysis for all non-significant results. We can’t claim anything other than what we have quantified. We feel the data of anosmic larvae are quite convincing because the behaviour changes so obviously.

(20) line 384. "Therefore…" Again, this implies the simulation result is telling you something, but you already knew this before simulating, based on the real anosmic data.

We have deleted the entire sentence.

(21) line 390-392, this seems like a really interesting idea -- could this be expanded/discussed further in the discussion part of the paper?

Thanks, we have expanded the idea in the discussion (see above).

(22) line 426ff. I think it would help to more clearly state which simulation results are obvious based on what you put in the simulation, and which gives you something unexpected.

We agree that it is a good idea to guide the reader in that sense. We have added a few comments:

Line 484: “As expected…”

Line 488: “Interestingly, despite the small differences we previously quantified, our results showed that sitter larvae consistently spent more time inside yeast patches than rovers for each number of patches (Figure 5—figure supplement 1D).”

Line 496: “to understand how fractioning environment would affect exploitation, which is key for survival.”

(23) line 480-482. Here too, larvae spending less time in less nutritious patches, isn't that a direct result of putting your empirical result into the simulation? They crawl faster and pause less, doesn't this have to be true?

The result of the simulation is indeed not surprising, but we can consider it as a sanity check for the model and for the choice of fitting parameters.

(24) line 636: is that hours after egg laying or after eclosion?

“Since egg laying” has been added.

(25) line 641ff: I don't quite follow -- you are choosing a lower frame rate in order to prioritize spatial resolution because the camera doesn't run faster when recording at 2048x2048?

It wasn’t clear. Line 767 “We recorded the movies at 2 frames per second to obtain forward movement displacements and actual pause-turns that are recorded accurately rather than to include ‘flickering’ movements associated with peristaltic movements.”

(26) line 690: when is instantaneous turn rate used? You are generally finding the total for long trajectories right?

We no longer use this phrase.

(27) line 695: What is the distance dimension parameter? Could you briefly explain what it means?

In a Douglas-Peucker algorithm “ε, represents the maximum distance between the original points and the simplified curve” added in line 829.

(28) line 699: typo with the brackets vs. parentheses.

Corrected, thanks!